# OneSearch: A Preliminary Exploration of the Unified End-to-End Generative Framework for E-commerce Search

**Ben Chen** [* 1]   **Xian Guo** [* 1]   **Siyuan Wang** [* 1]   **Zihan Liang** [* 1]   **Yufei Ma** [* 1]
**Yue Lv** [1]   **Chenyi Lei** [1]   **Yuqing Ding** [1]   **Wenwu Ou** [1]   **Han Li** [1]   **Kun Gai** [1]

## Abstract

Traditional e-commerce search systems employ multi-stage cascading architectures that suffer from fragmented computation and optimization objective collisions across stages, ultimately limiting their performance ceiling. We propose **One-Search**, the first industrial-deployed end-to-end generative framework for e-commerce search, featuring three key innovations: (1) Keyword-enhanced Hierarchical Quantization Encoding to preserve hierarchical semantics and distinctive item attributes while maintaining strong query-item relevance constraints; (2) multi-view user behavior sequence injection that constructs behavior-driven user IDs and incorporates both explicit short-term and implicit long-term sequences; and (3) a Preference-Aware Reward System with multi-stage supervised fine-tuning and adaptive reward-weighted ranking to capture fine-grained user preferences. Extensive offline evaluations demonstrate its superior performance, while online A/B tests achieve statistically significant improvements: +1.67% item CTR, +2.40% buyer, and +3.22% order volume. OneSearch reduces operational expenditure by 75.40%, improves Model FLOPs Utilization from 3.26% to 27.32%, and has been successfully deployed across multiple search scenarios in Kuaishou, serving millions of users daily. Code is in https://github.com/benchen4395/onesearch-family.

## 1. Introduction

E-commerce search systems aims to identify items satisfying both semantic and personalized criteria from hundreds

*Equal contribution   [1]Kuaishou Technology, Beijing, China. Correspondence to: Ben Chen <benchen4395@gmail.com>.

*Proceedings of the $43^{rd}$ International Conference on Machine Learning*, Seoul, South Korea. PMLR 306, 2026. Copyright 2026 by the author(s).

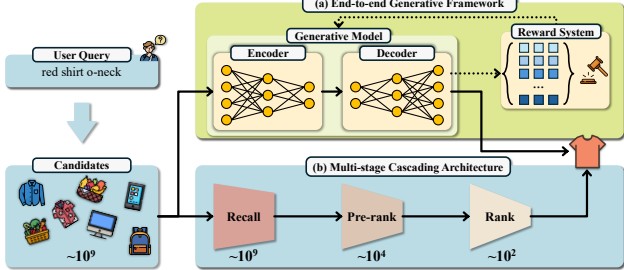

*Figure 1.* (a) Our proposed End-to-End generative retrieval framework, (b) the traditional multi-stage cascading architecture.

of millions of candidates within one second. Traditional systems employ Multi-stage Cascading Architecture (MCA), which progressively filters items through recall ($10^9$ candidates), pre-ranking ($10^4$ candidates), and ranking ($10^2$ candidates) stages, as shown in Figure 1(b).

While MCA effectively balances response time and accuracy, it suffers from two fundamental limitations (Grattafiori et al., 2024; Deng et al., 2025; Wei et al., 2025; Liang et al., 2025). First, fragmented compute: most serving resources are allocated to communication and storage rather than computation. Second, objective collision: heterogeneous optimization objectives across stages limit performance ceiling. Recall and pre-ranking stages prioritize coverage with lightweight models, while ranking emphasizes user preference reasoning with complex features. This multi-layer filtering risks discarding relevant items early, preventing optimal results regardless of subsequent model accuracy.

Recent efforts address these issues through either intra-stage optimization (Huang et al., 2020; Wang et al., 2021; Huang et al., 2013; Zhou et al., 2018; Guo et al., 2017) or cross-stage consistency (Song et al., 2022; Zhang et al., 2025; Evnine et al., 2024), yet remain constrained by MCA's inherent limitations. The emerging generative retrieval (GR) paradigm offers a promising alternative by transforming matching-based frameworks into generation-based approaches (Rajput et al., 2023; Zheng et al., 2024; Pang et al., 2025; Deng et al., 2025; Guo et al., 2026; Zheng et al., 2025; Wei et al., 2025; Zheng et al., 2026), eliminating multi-stage filtering through direct item generation.

However, e-commerce search presents challenges for GR adoption: (1) **Noisy item information**: sellers add irrelevant terms for exposure, creating lengthy descriptions with weak semantic that mislead representation models; (2) **Strict relevance constraints**: queries typically contain few keywords, where any attribute mismatch causes relevance issues. While semantic IDs provide hierarchical representations, they inevitably lose core attributes by prioritizing shared information; (3) **Latent intent inference**: uncovering user intent from concise queries requires effectively combining query content with behavior profiles. To address these issues, we propose **OneSearch**, an end-to-end generative framework for e-commerce search, which includes:

1) **K**eyword-enhanced **H**ierarchical **Q**uantization **E**ncoding (KHQE) module. It employs keyword-enhanced semantic collaborative encoding to highlight core item attributes, using RQ-Kmeans for hierarchical feature encoding and OPQ for unique feature quantization, further reducing noise and then enhancing query-item relevance.

2) **M**ulti-view **U**ser Behavior **Seq**uence (Mu-Seq) injection strategy. This strategy constructs behavior-driven user IDs with weighted decay sequences, explicitly incorporates short sequences for recent preferences, and implicitly models long sequences for comprehensive user profiles.

3) **P**reference **A**ware **R**eward **S**ystem (PARS). We implement multi-stage supervised fine-tuning (SFT) for semantic alignment and personalization, followed by adaptive reward-weighted ranking combining hierarchical behavior signals with list-wise preference optimization.

Extensive evaluations demonstrate OneSearch's superiority. Online A/B tests show statistically significant improvements: +1.67% item CTR, +2.40% buyer volume, and +3.22% order volume, while reducing operational expenditure by 75.40% and improving Model FLOPs Utilization from 3.26% to 27.32%. OneSearch is successfully deployed across multiple Kuaishou search scenarios, serving millions of users with tens of millions of daily PVs.

The main contributions are summarized as follows:

- A keyword-enhanced hierarchical quantization encoding balancing context features and collaborative signals while strengthening relevance constraints.

- A multi-view behavior sequence injection strategy, integrating user behavior sequences into ID representations and leveraging explicit/implicit prompts to enhance GRs' reasoning about user profiles and preferences.

- A preference aware reward system with multi-stage SFT and adaptive reward modeling for personalized ranking capability.

- Finally, we present **OneSearch**, the first industrial-deployed end-to-end generative framework for e-commerce search, validated by comprehensive offline and online experiments.

**Conflict of Interest Disclosure.** The authors Ben Chen, Xian Guo, Siyuan Wang, Zihan Liang, Yufei Ma, Yue Lv, Chenyi Lei, Yuqing Ding, Wenwu Ou, Han Li, and Kun Gai are employed by Kuaishou Technology, which leads the development of Online MCA and OneSearch, which were among the ones evaluated in this paper.

## 2. Related Works

In recent years, Generative Retrieval (GR) has garnered attention from both academia and industry due to its remarkable performance. Notable contributions in this area include Tiger (Rajput et al., 2023), DSI (Tay et al., 2022), and LC-REC (Zheng et al., 2024). Most GR models serve merely as supplementary recall sources within online systems, thereby overlooking these models' inherent rich semantic and powerful reasoning abilities for potential use in (pre-)ranking stages. In the area of video recommendation, OneRec (Deng et al., 2025) was the first to unify recall, pre-ranking, and ranking within a single generative model. Most advancements in generative retrieval have focused on recommendations. This is because search systems face three major challenges: 1) multiple and low-density item information, 2) strong relevance constraints between search queries and items, and 3) inference barriers to users' potential search intentions. Consequently, the current traditional e-commerce search systems still adopt a multi-stage cascading architecture. However, some efforts have been made to optimize current search systems using GR, e.g., GenR-PO (Li et al., 2024), GRAM (Pang et al., 2025) and OneSug (Guo et al., 2026). These GR methods demonstrate appealing performance in the realm of search, recommendation, bottom navigation, advertising, and query suggestion. Inspired by these works, we propose OneSearch, the first to unify all three stages (recall + pre-ranking + ranking) in e-commerce search. See extended discussion in Appendix A.

## 3. Methodology

### 3.1. Keyword-enhanced Hierarchical Quantization Encoding

Encoding items into Semantic IDs (SIDs) is crucial for generative retrieval models. This process converts continuous semantic representations into discrete ID sequences using coarse-to-fine quantization, ensuring items with the same SID share same information(Rajput et al., 2023; Deng et al., 2025; Ju et al., 2025). However, common quantization methods tend to tokenize shared signals using fixed vocabulary,

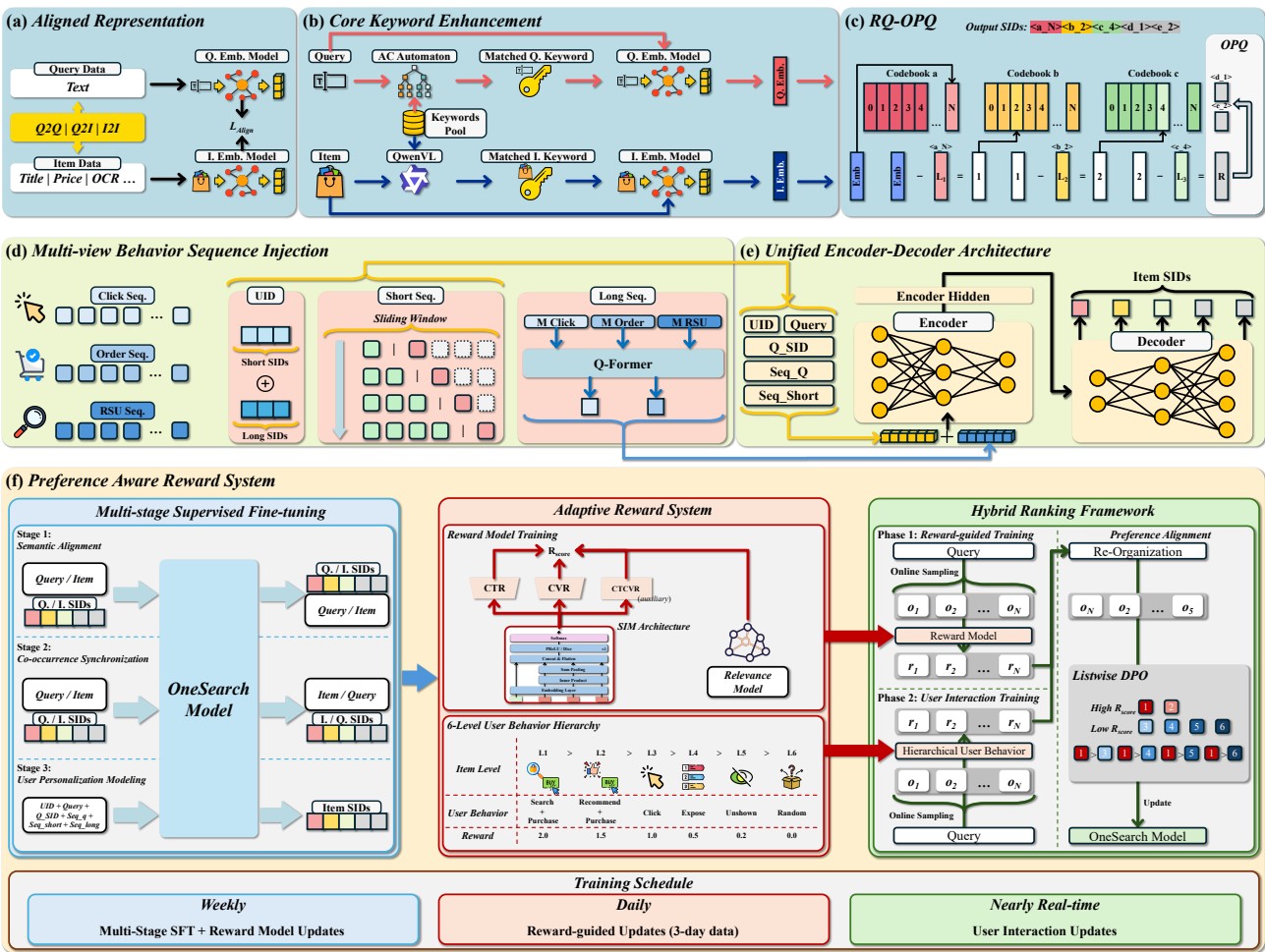

*Figure 2.* The Framework of OneSearch includes: 1) Keyword-enhanced Hierarchical Quantization Encoding, which adopts aligned representation and core keyword to construct a hierarchical quantization tokenization schema; 2) Multi-view Behavior Sequence Injection, utilizing sequences to reason the user profiles and preferences; 3) Unified encoder-decoder Architecture that integrates produced features for generative retrieval; and 4) Preference aware reward system, containing a multi-stage SFT procedure and an adaptive reward system to enhance the ranking capability.

losing distinctive attributes. We propose KHQE combining domain knowledge extraction, RQ-Kmeans for hierarchical encoding, and OPQ for unique feature quantization.

**Aligned collaborative and semantic representation.** We integrate semantic knowledge with collaborative signals by aligning the representations of historically interactive query-item pairs. Firstly, we select high-quality query2query, item2item, and query2item pairs from real user search logs using existing retrieval models like ItemCF (Sarwar et al., 2001) and Swing (Yang et al., 2020). Then we collect the content information like query text, item title, item price, keywords, OCR (image-to-text), as well as the statistical business characteristics, such as the number of clicks, add-to-cart, and purchases during a certain time. All these features are processed with a distilled BGE (Xiao et al., 2024) to generate a content embedding for each query $e_q$ and item $e_i$. Finally, we filter pairs with cosine similarity larger than

0.6 to ensure content relevance.

We design four interrelated tasks to align collaborative and semantic representation: 1) the query2query and item2item contrastive loss $\mathcal{L}_{q2q}$, $\mathcal{L}_{i2i}$ to align representations of collaboratively similar pairs, 2) a query2item contrastive loss $\mathcal{L}_{q2i}$ to ensure that BGE can reflect real business characteristics, 3) a query2item margin loss $\mathcal{L}_{rank}$ to further learn the collaborative signal deviation of query-item pairs with different behavior levels, 4) a hard sample relevance correction loss $\mathcal{L}_{rel}$, which uses MSE loss to fit LLM-scored relevance labels for hard query-item pairs with ambiguous similarity. Then we train the aligned model by the total loss with adjustable parameters $\lambda_i$ as:

$$\mathcal{L}_{align} = \lambda_1 \cdot \mathcal{L}_{q2q} + \lambda_2 \cdot \mathcal{L}_{i2i} + \lambda_3 \cdot \mathcal{L}_{q2i} + \lambda_4 \cdot \mathcal{L}_{rank} + \lambda_5 \cdot \mathcal{L}_{rel}, \quad (1)$$

**Core Keyword Enhancement.** Item textual information often contain redundant, irrelevant, or conflicting attributes.

Although boosting exposure, these disordered attributes makes it difficult for encoders to model key information. Here we utilize core keyword features to enhance textual representation, obtaining keyword-dominated semantic IDs.

Using named entity recognition (NER), we extract 18 structured attributes (detailed shown in Appendix B.1) and mine recent query-item pairs as labeled data. Core keywords are selected from PV-ranked attribute lists. Qwen-VL (Bai et al., 2023) identifies item keywords, while Aho-Corasick Automaton (Aho & Corasick, 1975) matches query keywords during inference. All these core keywords are fed into the trained model to obtain vectors $e_k^i$ consistent with the item representation distribution. The final optimized representations for each query $e_q^o$ and item $e_i^o$ are given by:

$$e_q^o = \frac{1}{2}(e_q + \frac{1}{m}\sum_{i=1}^{m} e_k^i), \quad e_i^o = \frac{1}{2}(e_i + \frac{1}{n}\sum_{j=1}^{n} e_k^j). \quad (2)$$

This keyword enhancement is specifically designed for e-commerce search, where item titles often contain noisy promotional text. This challenge is not in recommendation and can not be addressed by existing GR methods.

**RQ-OPQ Hierarchical Quantization Tokenization.** Common SID tokenizers like VQ-VAE (van den Oord et al., 2017), RQ-VAE (Lee et al., 2022), and RQ-Kmeans (Luo et al., 2025) prioritize encoding shared item features at the expense of distinctive characteristics, limiting generative GR performance. To address this limitation, we introduce a hybrid tokenizer called RQ-OPQ. We first employ RQ-Kmeans as the foundational tokenizer to capture hierarchical semantic structure. Then, we uniquely quantize the residual embedding discarded after RQ-Kmeans' final clustering step using OPQ (Ge et al., 2014). This approach allows RQ-Kmeans to model hierarchical relationships while OPQ simultaneously captures complementary lateral features. The resulting RQ-OPQ tokenizer provides a more comprehensive representation of fine-grained item characteristics, significantly strengthening the relevance constraints for downstream GR models. Implementation specifics and theoretical analysis on the information preservation properties of RQ-OPQ, are detailed in the Appendix B.

### 3.2. Multi-view Behavior Sequence Injection

**Behavior Sequence Constructed User IDs.** Random Hash IDs in Tiger (Rajput et al., 2023) do not adequately represent user personalization. Here, we propose a behavior sequence-constructed user ID for distinctive representation. Formally, the short behavior sequence consists of latest clicked items, denoted as $Seq_{short}$, length is $m$, and the long behavior sequence contains chronologically ordered items, denoted as $Seq_{long}$, length is $n$. The user ID $uid$ is computed as the concatenation of $SID_{short}$ and $SID_{long}$:

$$SID_{\text{short/long}} = \left\lceil \sum_{k\in\mathcal{K}} \gamma_k \cdot SID_k \right\rceil, \gamma_k = \frac{e^{\sqrt{k}}}{\sum_{k\in\mathcal{K}} e^{\sqrt{k}}} \quad (3)$$

where $\mathcal{K} = \{1, 2, \ldots, m\}/\{1, 2, \ldots, n\}$ denote the index of the short/long behavior sequence. For new-coming or cold-start users, we count the most clicked items for each query based on query-item occurrence and sort them in reverse order by page views as default behavior sequences.

**Explicit Short Behavior Sequence.** The short (recent) and long behavior sequences are crucial features in modeling user preferences. The short behavior sequence primarily reflects recent user preferences, while the long behavior sequence represents a user's profile. For example, a soon-to-be-enrolled college student may recently purchase items related to a new dormitory or their major. In contrast, purchases made six months ago might have been more related to college entrance exams or stationery. Therefore, for a user's next search, we prioritize the recent behavioral information. However, the long behavior sequence contains a user's long-term, stable preferences, such as a preference for cost-effectiveness, quality, or style. For the generative retrieval paradigm, explicitly inputting short behavior sequences makes it easier for the model to predict which categories of items users are most likely to click on.

For an e-commerce search platform, the short behavior sequences include the user's latest entered queries $Seq_q$ and clicked items $Seq_{short}$. We directly input the SIDs of these queries and items into the prompt, following the constructed user ID and the input query.

**Implicit Long Behavior Sequence.** For e-commerce platforms, user long behavior sequences primarily consist of three types: the click, the order, and the search relevant unit (RSU) (Guo et al., 2023) sequence, and the length of the behavior sequence is almost up to $10^3$. Therefore, it is almost impossible to integrate this information into the format of a handcrafted textual prompt. For each item within these sequences, we first map its keyword-enhanced embedding $e_i^o$ to a corresponding semantic ID (*sid* for simplicity), and then get RQ clustering centroid representation through the lookup method, which is considered to contain different levels of semantic information. Furthermore, we aggregate the centroids according to different levels, which not only allows the GR to systematically learn the user preferences at different levels, but also saves a lot of resources.

The overall pipeline is shown below, each item in the long-term historical sequence is replaced by the features of its RQ clustering centroid representative:

$$\mathbf{Item}_{sid} = \text{RQ}(e_i^o), \ \mathbf{Item}_{emb} = \text{EmbLookUp}(\mathbf{Item}_{sid}) \quad (4)$$

For long-term sequence, overall behavior embedding is

shown as:

$$\mathbf{M}_{c/o/r} = \left\{ \sum_{i=1}^{L1} \mathbf{Item}_{emb}^1, \sum_{i=1}^{L2} \mathbf{Item}_{emb}^2, \sum_{i=1}^{L3} \mathbf{Item}_{emb}^3 \right\},$$
$$\mathbf{Q}^{(i)} = \text{QFormer}(\mathbf{M}_c, \mathbf{M}_0, \mathbf{M}_r),$$
(5)

where $\mathbf{M}_c$, $\mathbf{M}_o$, and $\mathbf{M}_r$ are click / order / RSU sequence item embeddings, and share the same size $\mathbf{M} \in \mathbb{R}^{N_M \times d_{model}}$ ($d_{model} = 768$).

While user behavior modeling exists in recommendation systems, our explicit/implicit sequence separation specifically addresses the challenge of inferring search intent from concise queries (typically 2-3 keywords), which differs fundamentally from recommendation.

### 3.3. Unified Encoder-Decoder Architecture

The input of OneSearch contains: 1) Distinctive user ID $uid$. 2) Entered query $q$ and its $SID_q$; 3) User short behavior sequence, containing the historical search queries $Seq_q = \{q_1, q_2, \ldots, q_n\}$, the short clicked item sequence $Seq_{short} = \{s_1, s_2, \ldots, s_n\}$; 4) Implicit long behavior sequence $Seq_{long}^{emb}$; 5) User profile information $\mathcal{U}$, which is the crowd portrait fitted by the platform. OneSearch directly outputs corresponding item lists $\mathcal{I}$. The model adopts either encoder-decoder models (e.g. BART (Lewis et al., 2020), mT5 (Xue et al., 2020)), or the decoder-only models (e.g. Qwen3 (Yang et al., 2025)) as the backbone $\mathcal{M}$. The inference flow can be formalized as:

$$\mathcal{I} := \mathcal{M}(uid, q, SID_q, Seq_q, Seq_{short}, Seq_{long}^{emb}, \mathcal{U}). \quad (6)$$

As illustrated in Figure 2, our model adheres to the transformer-based (Vaswani et al., 2017) architecture, comprising an encoder that models $\langle user, query, seq \rangle$ information, and a decoder dedicated to item generation. We adopt encoder-decoder models for deployment due to architecturally accelerated training and inference. For unified training, we insert $t_{[BOS]}$ and $t_{[EOS]}$ at boundaries, with $t_{[SEP]}$ between adjacent elements. The inference output of $\mathcal{M}$ is the SIDs, which can be adjusted by constrained or unconstrained beam search. While constrained search guides output to valid SIDs, it increases decoding complexity, and unconstrained explores all sequences without explicit rules.

### 3.4. Preference Aware Reward System

Compared to the sequence coherence in recommendation systems, strong relevance constraints between queries and items in search engines pose greater challenges for online MCA. For GR models, it is necessary not only to achieve semantic alignment between SIDs and the textual descriptions of queries and items but also to directly generate items that meet query relevance constraints and user preferences based on historical behavior sequences. Therefore, we propose a preference aware reward system, which includes a

multi-stage supervised fine-tuning (SFT) and an adaptive reward system, to enhance the model's personalized ranking capability, which is depicted in Figure 2(f).

**Multi-stage Supervised Fine-tuning.** onsidering that the basic architecture (e.g., BART, T5) is pretrained with a large text corpus, but the input queries and items in OneSearch are represented using SIDs, it is essential to first achieve semantic alignment between SIDs and their corresponding textual descriptions. Subsequently, the model should be instructed to generate the desired items that align with user intentions. To address this, we have designed a multi-stage supervised fine-tuning (SFT) procedure.

1. **Semantic Content Alignment**: Three sub-tasks: (a) Query/Item Text→SID generation, (b) SID→text reconstruction, (c) Text/item→category prediction. The first two tasks align SID and text content, while category prediction ensures relevance.

2. **Co-occurrence Synchronization**: Mutual prediction between query↔item and query SID↔item SID. Without user profile, this stage learns intrinsic semantics and collaborative relationships from interactive corpus.

3. **User Personalization Modeling**: After the above stages, we introduce user information aligning with online inference. We concatenate user ID, query, $SID_q$, $Seq_q$, $Seq_{short}$, and $Seq_{long}^{Emb}$ as input, with item SID as the training label for distinctive personalization.

We apply sliding window data augmentation to short sequences to guide learning of user interest changes. The sliding window generates new segments with subsequent items as prediction targets by sliding along $Seq_{short}$ (Zhou et al., 2024). With maximum window length limitation, we augment $m$ samples for $Seq_{short} = \{s_1, s_2, \ldots, s_m\}$, helping handle new users with limited history.

**Adaptive Reward System.** Unlike OneRec-V1 (Zhou et al., 2025a), which uses a weighted P-Score of multiple objectives to train one reward model followed by Early Clipped GRPO to guide the model in learning user preferences, we use real online user interactions as feedback signals. These interactions provide more accessible and hierarchical feedback information. While this approach shares similarities with the recently proposed OneRec-V2 (Zhou et al., 2025b), there are significant differences in training data sampling and training paradigms. We adopt adaptive-weighted reward signals (Guo et al., 2026) to construct training data and implement a user-behavior-guided hybrid ranking framework to achieve personalized preference ranking.

**Adaptive-weighted Reward Signal.** Following OneSug (Guo et al., 2026) we categorize interactive behaviors into

*Table 1.* The preference-aware reward system combines a three-stage fine-tuning process (semantic alignment, co-occurrence, personalization) and an adaptive ranking mechanism.

| Procedure | SFT Stage 1 | SFT Stage 2 | SFT Stage 3 | RL Stage |
|---|---|---|---|---|
| **Objective** | Semantic alignment | $\langle q, i \rangle$ co-occurrence | User personalization | Preference Alignment |
| **Component** | query $\leftrightarrow$ SID
item $\leftrightarrow$ SID
query/item $\mapsto$ category
SID $\mapsto$ category | query $\leftrightarrow$ item
query_SID $\leftrightarrow$ item_SID | $\begin{bmatrix} uid\ \&\ q \\ SID_q\ \&\ Seq_q \\ Seq_{short} \\ Seq_{long}^{emb} \end{bmatrix} \mapsto$ item_SID | $\begin{bmatrix} \text{user \& query} \\ \text{seq. feat.} \\ item_{win} \\ item_{lose} \end{bmatrix} \mapsto$ Rank Score |

six levels: (1) purchased in search, (2) same-category purchased in recommendation, (3) clicked, (4) exposed-not-clicked, (5) unshown same-category, (6) random other-category. Base weights are $\lambda = [2.0, 1.5, 1.0, 0.5, 0.2, 0.0]$. Considering items with higher recent CTR/CVR are more likely to be selected, we utilize these two metrics to construct adaptive-weighted rewards. However, CTR and CVR often suffer from biased estimation. For example, a newly released item that was exposed only once and then clicked would have a CTR of 100%. Conversely, most popular items are often exposed by Online MCA under various similar but suboptimal queries, which results in lower CTR and CVR.

Therefore, we calibrate these two metrics as follows:

$$Cnt_T = \log((Cnt_{pos} + 10) \cdot (Cnt_c + 10) \cdot (Cnt_o + 10)).$$
$$Ctr_i = \frac{\log(Cnt_c + 10)}{Cnt_T}, \ Cvr_i = \frac{\log(Cnt_o + 10)}{\log(Cnt_c + 10)}. \quad (7)$$

The weighted reward score is then defined as:

$$r(q, i) = 2\lambda \cdot \frac{Ctr_i \cdot Cvr_i}{Ctr_i + Cvr_i}. \quad (8)$$

For each positive sample $i_{pos}$ and negative sample $i_{neg}$, the user preference difference $rw_\Delta$ is:

$$rw_\Delta = 1.0 \ / \ [r(q, i_{pos}) - r(q, i_{neg})], \quad (9)$$

where smaller $rw_\Delta$ encourage the model to distinguish nuanced differences in user behaviors.

**Reward Model Training.** As discussed in OneRec-V2(Zhou et al., 2025b), the reward model in OneRec-V1 employs restricted sampling from a small subset of users to approximate global behavior, potentially learning specific patterns or biases that do not yield actual improvements. However, we also diverge from the feedback-driven preference alignment proposed in OneRec-V2, as the adoption of GRPO and its variants (e.g., ECPO, GBPO) tends to introduce more irrelevant SIDs, and preference rewards require careful tuning for e-commerce search. Here we design an intuitive three-tower SIM (Pi et al., 2020)) architecture, with each tower learning CTR, CVR, CTCVR (Ma et al., 2018) using binary cross-entropy. The preference score is:

$$Rscore = \lambda_1 \cdot Ctr + \lambda_2 \cdot Cvr + \lambda_3 \cdot Ctcvr + 10 \cdot \lambda_4 \cdot S_{Rel}, \quad (10)$$

where $\lambda_i$ represents tuned weights. To ensure that the results generated by OneSearch meet relevance constraints, we additionally incorporate an offline-calculated relevance score $S_{Rel}$ with an amplified weight ($10 \cdot \lambda_4$).

This reward model differs from the click prediction model in the ranking stage of online MCA in two key aspects: (1) Feature dimensionality: While the ranking model utilizes thousands of features, our reward model only takes user ID, query, user behavior sequence, and user profile as input, matching OneSearch's input space. (2) Sampling strategy: We additionally include items from the same category clicked in recommendation scenarios as training samples, with labels (1,1,1) for purchased items and (1,0,0) for clicked items. For computational efficiency, the reward model directly leverage the online MCA ranking model, as we only distill the ranking order rather than absolute scores. Note that the RM is used only during training; the deployed system is the pure generative model without any reranking.

**Hybird Ranking Framework.** The alignment stage comprises two components. First, we collect entered queries from real search logs and use the reward model to rerank items output by the fine-tuned OneSearch. We then select samples where ranking changes occur for list-wise DPO training. Items that are clicked or advanced in position by the reward score serve as positive samples, while items pushed back in ranking and those at lower positions serve as negative samples. This allows us to gather one positive sample and multiple negative samples per query for training. The optimization objective can be described as follows:

$$\mathcal{L} = -\mathbb{E}\Bigg[\log \sigma\bigg(\log \sum_{i_l \in \mathcal{I}_l} \exp\Big(rw_\Delta \max\big(0, \hat{r}_\theta(x_u, i_w)$$
$$- \hat{r}_\theta(x_u, i_l) - \delta\big)\Big)\bigg) + \alpha \log \pi_\theta(i_w | x_u)\Bigg], \quad (11)$$

where $\mathcal{I}_l$ denotes the set of negative samples, and $\hat{r}_\theta(x_u, i_w)$ and $\hat{r}_\theta(x_u, i_l)$ represent rewards implicitly defined by the language model $\pi_\theta$ and reference model $\pi_{\text{ref}}$:

$$\hat{r}_\theta(x_u, i_{w/l}) = \beta \log \frac{\pi_\theta(i_{w/l} | x_u)}{\pi_{\text{ref}}(i_{w/l} | x_u)}. \quad (12)$$

The term $\log \pi_\theta(i_w | x_u)$ represents the log-likelihood (NLL loss) from the SFT stage. Noted that by combining the list-wise preference alignment with log-likelihood prediction of

preferred samples, we establish a novel hybrid paradigm for generative ranking. Since the reward model trains on MCA data, it inherently limits OneSearch's performance. Thus, phase two uses pure user interactions: positives from top behaviors (purchased, same-category purchased, clicked) and negatives from bottom behaviors (exposed-not-clicked, unshown same-category, random), with the same loss. In practice, we periodically perform first-phase RL with reward model-generated samples to ensure online distribution adherence and learn from the MCA ranking model's thousands of features. The second phase updates near-streaming with user interaction data to overcome distribution limitations and fully leverage generative inference capabilities. This dual approach enables OneSearch to benefit from both MCA's rich feature space and direct user feedback.

## 4. Experiment

In this section, we conduct comprehensive evaluations on practical industry datasets and online A/B tests to verify the feasibility of OneSearch, followed by ablation studies. We note that no public e-commerce search dataset includes complete user behavior logs with profiles, queries, and multi-level interactions. We commit to releasing our evaluation dataset under company limitations.

**Datasets.** We extract the user interactive pairs from Kuaishou's mall search platform between May 2025 and August 2025 to facilitate the supervised fine-tuning and DPO. It contains about 1 billion PVs, all the offline and ablation experiments were conducted on the full or part of this data.

**Evaluation Metrics.** We employ HitRate (HR) and Mean Reciprocal Ranking (MRR) for recall and ranking performance, which are standard metrics in search and recommendation systems. We report HR@n and MRR@n, where n=350 corresponds to the 99th-percentile of items viewed per search session in our online system, representing the actual user experience boundary and ensuring fair comparison across all pipeline variants.

**Baselines.** We compare OneSearch against OnlineMCA, our production Multi-stage Cascading Architecture that progressively filters items through recall ($10^9 \rightarrow 10^4$), pre-ranking ($10^4 \rightarrow 10^2$), and ranking stages. Each module in OnlineMCA combines multiple SOTA models with thousands of features. This comparison directly measures whether OneSearch can replace the full production pipeline. Existing GR methods only optimize partial stages and are not directly comparable.

### 4.1. Offline Performance

We evaluated 30,000 click pairs and 30,000 order pairs from user search logs, computing HR@350 and MRR@350. Table 2 shows that pre-ranking prioritizes recall over precision

*Table 2.* Offline performances of our proposed method with onlineMCA on the industry dataset. The best results are in bold, and sub-optimal results are underlined in each column. The "w/o ranking" means "without ranking", and the "\+ keywords " means "add keywords optimizations"

| Method | order (30k) | | click (30k) | |
|---|---|---|---|---|
| | **HR@350** | **MRR@350** | **HR@350** | **MRR@350** |
| OnlineMCA | 51.74% | 19.26% | 64.40% | 16.89% |
| w/o ranking | 75.75% | 4.19% | 80.23% | 3.00% |
| OPQ (8/256) | 19.43% | 9.55% | 22.57% | 7.42% |
| (1024-1024-1024) | 57.39% | 9.12% | 63.63% | 7.46% |
| (2048-1024-512) | 58.29% | 10.79% | 65.39% | 8.86% |
| (4096-1024-256) | 58.57% | 11.21% | 64.51% | 9.24% |
| (4096-1024-512) | 59.58% | 14.29% | 62.49% | 11.82% |
| \+ keywords | 62.38% | 14.30% | 66.14% | 12.10% |
| \+ l3 balanced | 63.16% | 13.59% | 68.26% | 11.67% |
| \+ Adaptive RS | 64.33% | 16.11% | 68.94% | 13.80% |
| RQ-OPQ (2/256) | 65.05% | 15.33% | 68.88% | 12.90% |
| \+ Adaptive RS | **66.46%** | **18.38%** | **71.06%** | **16.33%** |

(75.75% HR but 4.19% MRR for orders), while ranking emphasizes intent positioning. This demonstrates MCA's optimization objective collision—final ranking is constrained by pre-ranking outputs. Note that while 'w/o ranking' achieves highest HR@350 (75.75%), its MRR@350 is significantly lower (4.19%), indicating poor personalized ranking. **OneSearch** balances both metrics: achieving superior recall over full OnlineMCA while maintaining comparable ranking precision, which is critical for user experience.

Testing various RQ-Kmeans and KHQE configurations (Table 6), we found higher codebook utilization rate (CUR) and independent coding rate (ICR) improve performance. Core keyword enhancement and L3 balanced k-means both provide improvements. Besides, adaptive reward preference learning significantly enhances ranking (+1.80% HR@350, +3.24% MRR@350 average).

Our final configuration, RQ-OPQ (2/256) with Adaptive RS, combines RQ-Kmeans (4096-1024-512) for hierarchical encoding with OPQ (256-256) for residual quantization, trained using the full preference aware reward system. This achieves superior recall (66.46% vs. 51.74% for orders) and comparable MRR (18.38% vs. 19.26%) to onlineMCA, effectively balancing personalized ranking with intent-matching. The configuration maintains robust performance across both click and order metrics, validating our hybrid tokenization approach. This configuration would be called OneSearch in the following section for brevity. Additional implementation details are in Appendix C.

### 4.2. Ablation Study

**Multi-view Behavior Sequences.** Table 3 demonstrates each component's contribution. Sequence-constructed user IDs outperform hashing IDs (+1.33% HR@350). Short sequences provide the largest gains (+3.79% HR@350,

*Table 3.* Ablation study of multi-view behavior sequence injection. Slid. Window means the sliding window strategy.

| Method | order (30k) | | click (30k) | |
| --- | --- | --- | --- | --- |
| | HR@350 | MRR@350 | HR@350 | MRR@350 |
| OneSearch | 66.46% | 18.38% | 71.06% | 16.33% |
| w/o User SIDs | -0.94% | -0.37% | -1.72% | -0.36% |
| w/o $Seq_{short}$ | -3.43% | -1.53% | -4.15% | -1.32% |
| w/o $Seq_{long}^{emb}$ | -2.26% | -1.01% | -3.00% | -1.05% |
| w/o Slid.Window | -1.95% | -0.81% | -1.80% | -0.70% |

*Table 4.* Online results for A/B testing. The black fonts indicate that the statistical significance (P-value) is smaller than 0.05, while the gray ones are larger than 0.05 indicating low confidence.

| Method | Item CTR | PV CTR | PV CVR | Buyer | Order |
| --- | --- | --- | --- | --- | --- |
| MCA w/o ranking | -9.97% | -20.33% | -11.55% | -28.78% | -39.14% |
| OneSearch[1] | -1.10% | -2.06% | +0.39% | +1.27% | -2.22% |
| OneSearch[1]$_{RM}$ | +1.40% | +3.05% | +1.94% | +1.92% | +1.59% |
| OneSearch[2] | +1.45% | +1.40% | -0.12% | -0.58% | -0.69% |
| OneSearch[2]$_{RM}$ | **+1.67%** | **+3.14%** | **+1.78%** | **+2.40%** | **+3.22%** |

+1.43% MRR@350), validating explicit preference modeling. Long sequences and sliding window augmentation further enhance performance.

**Tokenization Stability.** E-commerce inventory changes constantly, especially during shopping festivals, potentially disrupting pre-calculated SID pools. We tested tokenizer stability by constructing RQ-Kmeans and RQ-OPQ using July 15 items and tracking performance through August 18 promotions. Figure 3 shows minimal degradation: RQ-Kmeans decreased 1.11% in CUR while RQ-OPQ only 0.43%, validating RQ-OPQ's superior robustness to inventory changes. Detailed RQ-OPQ ablations are in Appendix D.

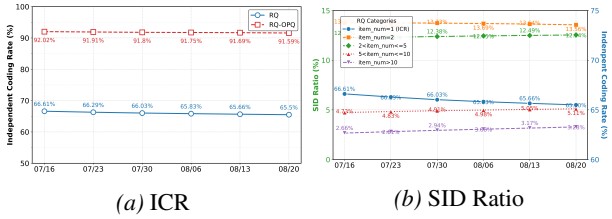

*(a)* ICR       *(b)* SID Ratio

*Figure 3.* The ICR and SID ratio indicators of RQ-Kmeans after regular time intervals.

### 4.3. Online A/B Testing

We conducted rigorous A/B tests on Kuaishou's mall search platform. Table 4 presents results for two OneSearch versions: OneSearch[1] (RQ-Kmeans without long sequences) and OneSearch[2] (full optimizations), each tested with and without reward model reranking ($RM$).

Table 4 demonstrates that the base generative model achieves comparable performance to the complex MCA

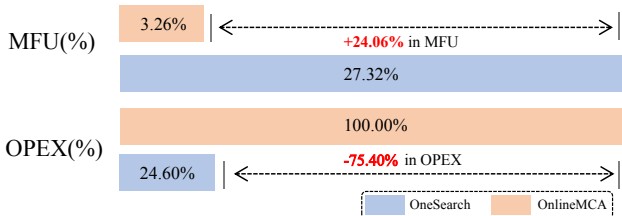

*Figure 4.* The comparisons of MFU and OPEX for onlineMCA and OneSearch.

system. With RQ-OPQ and long behavior sequences, OneSearch[2] improves item CTR by 1.45% and PV CTR by 1.40%. Incorporating reward model selection (OneSearch[2]$_{RM}$) yields statistically significant gains across all metrics: +1.67% item CTR, +3.14% PV CTR, +1.78% PV CVR, +2.40% buyer volume, and +3.22% order volume.

For a clearer comparison, we perform additional experiments on the online search system, named MCA w/o ranking, which only uses the "recall and pre-ranking" module to predict the items, without the ranking stage. It significantly reduces all indicators, especially with 28.78% in Buyer, 39.14% in Order volume. This indirectly verifies that OneSearch has comparable ranking capabilities. These outstanding results show that OneSearch outperforms the onlineMCA and indicate it can update the complicated online system to a more balanced state without generating seesaw effects. Last but not least, we also observe improvements in manual evaluations, detailed manual evaluation results are provided in Appendix E. Ultimately, OneSearch has been successfully deployed across the e-commerce search ecosystem, including full traffic on the detail page search, 50% of traffic on the mall search, and 20% of traffic on the homepage search for online evaluation. The system currently serves millions of users and millions of page views daily.

### 4.4. Further Analysis

**Computational Efficiency.** We measured Model FLOPs Utilization (MFU) on flagship GPUs during serving inference. As shown in Figure 4, OneSearch achieves 27.32% MFU compared to onlineMCA's 3.26%, an 8.4× improvement. This significantly outperforms traditional MCA systems while approaching the ~40% MFU typical of LLMs on H100 GPUs (Grattafiori et al., 2024). Furthermore, OneSearch reduces operational expenditure (OPEX) to 24.60% of the online search pipeline by eliminating communication and memory overhead between stages. Additional discussion is provided in the Appendix F.

### 5. Conclusion

In this paper, we present OneSearch, an end-to-end generative framework for e-commerce search system that effec-

tively overcomes the limitations of traditional multi-stage cascading architectures. By employing a unified generative model, introducing keyword-enhanced hierarchical quantization encoding, multi-view behavior sequences injection, preference aware reward system, OneSearch achieves superior semantic understanding and personalization modeling. Extensive offline and online evaluations confirm OneSearch's effectiveness in boosting CTR and business conversions. OneSearch sets a new benchmark for industrial search solutions, paving the way for future advancements in generative retrieval methods.

## Impact Statement

This research uses anonymized user interaction data from an e-commerce platform in compliance with privacy policies and data protection regulations. The deployment affects millions of users, and we have implemented monitoring systems to detect potential biases or adverse effects. A/B testing methodology ensures gradual rollout to minimize user risks. We recognize algorithmic search systems can influence user behavior and have taken measures to promote relevant, diverse results across product categories. The research balances commercial objectives with user value, and the system continues to be monitored for ethical implications.

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

## A. Related Works

**Generative Retrieval and Recommendation.** In recent years, Generative Retrieval (GR) has garnered significant attention from both academia and industry due to its remarkable performance. This emerging retrieval paradigm, which regards large-scale retrieval as sequence-to-sequence generation tasks, has outperformed traditional ANN-based models such as EBR (Huang et al., 2020) and RocketQA (Qu et al., 2021), spurring increased exploration in the fields of search and recommendation. Notable contributions in this area include Tiger (Rajput et al., 2023), DSI (Tay et al., 2022), and LC-REC (Zheng et al., 2024). Tiger (Rajput et al., 2023) pioneered the development of end-to-end generative retrieval models for sequential recommendation, introducing semantic IDs (SID) derived from each item's content information for efficient item representation. LC-REC (Zheng et al., 2024) proposed adapting large language models (LLMs) (Liang et al., 2024; Ma et al., 2024; Wang et al., 2025) by integrating collaborative semantics for recommendation, utilizing a series of specially designed tuning tasks.

Most GR models serve merely as supplementary recall sources within online systems, thereby overlooking these models' inherent rich semantic and powerful reasoning abilities for potential use in (pre-)ranking stages. In the area of video recommendation, OneRec (Deng et al., 2025) was the first to unify recall, pre-ranking, and ranking within a single generative model. This was achieved with the assistance of session-wise generation and iterative preference alignment, resulting in substantial improvements in practical online metrics. EGA (Zheng et al., 2025) represents a significant departure from both traditional multi-stage cascading architectures (MCA) and existing generative retrieval models by introducing a unified framework that holistically models the entire advertising pipeline. UniROM (Qiu et al., 2025) employs a hybrid feature service to efficiently decouple user and advertising features, and RecFormer (Li et al., 2023), a variation of Transformer, captures both intra- and cross-sequence interactions.

**Generative Retrieval for Search.** These two years, most advancements in generative retrieval have been focused on recommendations. This is because search systems face three major challenges: 1) multiple and low-density item information, 2) strong relevance constraints between search queries and items, and 3) inference barriers to users' potential search intentions. Consequently, the current traditional e-commerce search systems still adopt a multi-stage cascading architecture. However, some efforts have been made to optimize current search systems using generative retrieval (GR).

The first example is GenR-PO (Li et al., 2024), which utilizes multi-span identifiers to represent raw item titles. This approach transforms the task of generating titles from queries into the task of generating multi-span identifiers from queries, thereby simplifying the generation process. Subsequently, a constrained search method is employed to identify key spans for retrieving the final item, which has proven beneficial for online recall systems. Another notable example is the Generative Retrieval and Alignment Model (GRAM) (Pang et al., 2025), which performs joint training on text information from both queries and products to generate shared text identifier codes. GRAM employs a co-alignment strategy to optimize these codes for maximizing retrieval efficiency and is deployed on the JD search engine to enhance both the recall and pre-ranking stages. The OneSug (Guo et al., 2026) in query suggestion, which incorporates a prefix2query representation enhancement module to enrich prefixes using semantically and interactively related queries to bridge content and business characteristics, an encoder-decoder generative model that unifies the query suggestion process, and a reward-weighted ranking strategy with behavior-level weights to capture fine-grained user preferences. It is the first end-to-end generative framework for e-commerce query suggestion, and has been verified to have substantial improvements in user clicks and conversion.

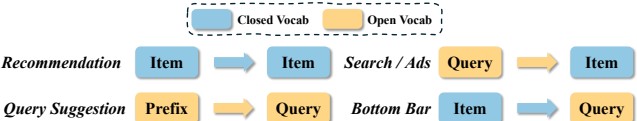

*Figure 5.* The input and output differences among Recommend, Search/Ads, Query Sug and Bottom Bar.

These GR methods demonstrate appealing performance in the realm of search, recommendation, bottom navigation, advertising, and even query suggestion. They are not suitable for e-commerce search. As illustrated in Figure 5, the inputs and outputs of recommendation are the closed-vocabulary items or videos, thus the pure semantic ID tokenization is suitable for its diverse item generation. The inputs and outputs of query suggestion are the full open-vocabulary textual descriptions, so that it can directly use the transformer architecture. For the bottom bar and search engine, either the inputs or the outputs are open-vocabulary, which represents a significant departure from both OneRec and EGA.

# B. Keyword-enhanced Hierarchical Quantization Encoding Details

## B.1. Core Keyword Enhancement Details

To enhance the role of core keywords in encoding, the core keyword enhancement scheme introduce 18 structured attributes (shown in Table 5) that improves the codebook utilization rate (CUR) of RQ-Kmeans at each level and further increases the independent coding rate (ICR). For example, with a configuration of 4096-1024-512 in Table 6, it results in a 0.10% CUR increment for Level 1, 24.84% for Level 2, and 26.15% for Level 3, as well as the overall ICR increasing by 6.86%.

*Table 5.* 18 structured attributes using NER in Kuaishou e-commerce search platform.

| Attribute Types | | | | | | | | |
|---|---|---|---|---|---|---|---|---|
| Entity Scene | Modifier Specifications | Brand Price | Material Model | Style Anchor | Function Series | Location Marketing | Audience Season | Color Pattern |

*Table 6.* The codebook utilization rate (CUR), which is the number of non-empty codebooks divided by the size of codebooks, and independent coding rate (ICR), which is the percentage of non-empty codebooks in total layer that contain only one item, for various RQ-Kmeans configurations. The last $^+$ means balanced operation for all levels.

| Configurations | $\text{CUR}_{L1}$ | $\text{CUR}_{L1*L2}$ | $\text{CUR}_{Total}$ | ICR |
|---|---|---|---|---|
| 1024-1024-1024 | 100% | 54.27% | 1.72% | 36.67% |
| \+keywords | 100% | 65.40% | 2.03% | 40.25% |
| 2048-1024-512 | 100% | 46.88% | 1.98% | 37.80% |
| \+keywords | 100% | 57.16% | 2.51% | 40.76% |
| 4096-1024-256 | 99.90% | 39.21% | 2.27% | 36.98% |
| \+keywords | 100% | 48.95% | 2.94% | 40.52% |
| \+l3 balanced | 100% | 48.95% | 10.31% | 60.01% |
| 4096-1024-512 | 99.90% | 39.21% | 1.30% | 40.54% |
| \+keywords | 100% | 48.95% | 1.64% | 43.32% |
| \+l3 balanced | 100% | 48.95% | 7.03% | 68.08% |
| 4096-1024-512$^+$ | 99.93% | 41.45% | 0.51% | 33.47% |

*Table 7.* Performance comparisons of three tokenization schemas evaluated on the real click pairs.

| Method | $\text{CUR}_{Total}$ | ICR | Recall@10 | MRR@10 |
|---|---|---|---|---|
| OnlineMCA | - | - | 0.3440 | 0.1323 |
| RQ-VAE | 1.17% | 38.83% | 0.2171 | 0.0689 |
| RQ-Kmeans | 7.03% | 68.08% | 0.2844 | 0.1038 |
| RQ-OPQ | - | **91.91%** | **0.3369** | **0.1194** |

## B.2. RQ-OPQ tokenization Details

Here, we use CUR and ICR as evaluation metrics. The basic codebook size is set to 1024, the number of codebook layers is set to 3, and the size of codebooks can be $10^9$, which aligns with the number of items in the candidate pool. However, e-commerce items have more varied categories and attributes, and RQ-Kmeans tends to prioritize clustering shared prominent features in the former layers. In order to make more concise tokenization, we maintain the capacity of codebooks while increasing the width of the former layer to ensure more comprehensive learning of prominent features. As depicted in Table 6, we tested three configurations: (1024-1024-1024), (2048-1024-512), and (4096-1024-256). The set of (4096-1024-256) achieves higher CUR and ICR, and the core keyword enhancement scheme (\+keywords) shows further improvement. Considering that the search system should encode the entered query similarly and that merchants often increase the number of listed items during global shopping festivals (e.g., 11.11 and 6.18), we further expanded the codebook size to (4096-1024-512). We found that the semantic tokens increased by 11.56% (as $2 \cdot 1.64\%/2.94\% - 1$), and the ICR increased to 43.42% compared to the (4096-1024-512) (\+keywords).

To further improve CUR and ICR, OneRec-V1 (Deng et al., 2025) proposed using full layers balanced k-means. However, for complex fine-grained attributes of items, forcing them into the same cluster in the early stages can lead to hierarchical

clustering collapse. As shown in Table 6, the $\text{CUR}_{Total}$ for the balanced k-means operation on full layers (4096-1024-512$^+$) is much lower than the (\+keywords) configuration. The CUR drastically decreased from 48.95% of $\text{CUR}_{L1+L2}$ to 1.64% in $\text{CUR}_{Total}$, indicating that many similar items were assigned the same ID. Therefore, we propose applying balanced k-means only to the codebook of the third layer to achieve independent encoding of similar items. As shown in (\+l3 balanced), the $\text{CUR}_{Total}$ increased from 1.64% to 7.03%, while the ICR improved by 57.16%.

Although RQ-Kmeans can construct hierarchical, learnable SIDs for items, it inevitably discards the residual embedding computed after the last clustering. However, this residual embedding contains the distinctive attributes of each item. Therefore, we further use OPQ for quantizing the unique features. The RQ method handles hierarchical semantics, while OPQ is adopted for lateral characteristics. This combined tokenizer can more comprehensively represent the fine-grained features of items, thereby enhancing the relevance constraints for GR models. As shown in Table 7, the two additional SIDs (256-256) generated by OPQ significantly improve the ICR metric and enhance the recall and ranking capabilities of GRs. A detailed ablation study on OPQ tokenization is extended in §D.

## B.3. Theoretical Analysis of RQ-OPQ Information Preservation

We provide theoretical analysis for the information preservation properties of the RQ-OPQ hierarchical quantization encoding.

### B.3.1. PRELIMINARIES AND PROBLEM SETUP

Let $\mathbf{x} \in \mathbb{R}^d$ denote the keyword-enhanced item embedding drawn from a data distribution $\mathcal{P}$ supported on a bounded set $\mathcal{X} \subset \mathbb{R}^d$. The RQ-OPQ tokenization scheme consists of two components:

1. **RQ-Kmeans**: A residual quantization with $L$ layers and codebooks $\{\mathcal{C}^{(l)}\}_{l=1}^{L}$ where $\mathcal{C}^{(l)} = \{c_1^{(l)}, \ldots, c_{K_l}^{(l)}\} \subset \mathbb{R}^d$ and $K_l \in \mathbb{N}^+$ denotes the codebook size at layer $l$.

2. **OPQ**: Optimized Product Quantization applied to the residual after RQ-Kmeans, with $M$ subspaces (where $M$ divides $d$) and sub-codebooks $\mathcal{D} = \{\mathcal{D}^{(m)}\}_{m=1}^{M}$, each satisfying $\mathcal{D}^{(m)} \subset \mathbb{R}^{d/M}$ with $|\mathcal{D}^{(m)}| = K_{\text{OPQ}}$.

**Definition B.1** (RQ-Kmeans Quantization). Given input $\mathbf{x} \in \mathbb{R}^d$, define the initial residual as $r^{(0)}(\mathbf{x}) := \mathbf{x}$. The RQ-Kmeans quantization recursively computes for $l = 1, \ldots, L$:

$$q^{(l)}(\mathbf{x}) := \arg \min_{k \in [K_l]} \|r^{(l-1)}(\mathbf{x}) - c_k^{(l)}\|_2, \tag{13}$$

$$r^{(l)}(\mathbf{x}) := r^{(l-1)}(\mathbf{x}) - c_{q^{(l)}(\mathbf{x})}^{(l)}, \tag{14}$$

where $[K_l] := \{1, 2, \ldots, K_l\}$. The reconstructed embedding is:

$$\hat{\mathbf{x}}_{\text{RQ}} := \sum_{l=1}^{L} c_{q^{(l)}(\mathbf{x})}^{(l)}. \tag{15}$$

**Definition B.2** (Optimized Product Quantization (OPQ)). Let $R \in \mathbb{R}^{d \times d}$ be an orthogonal matrix. For input $\mathbf{z} \in \mathbb{R}^d$, OPQ performs:

1. **Rotation**: Compute $\mathbf{y} := R^\top \mathbf{z} \in \mathbb{R}^d$.

2. **Partition**: Decompose $\mathbf{y} = [\mathbf{y}^{(1)}; \ldots; \mathbf{y}^{(M)}]$ where $\mathbf{y}^{(m)} \in \mathbb{R}^{d/M}$.

3. **Quantization**: For each $m \in [M]$:

$$d^{(m)}(\mathbf{z}) := \arg \min_{k \in [K_{\text{OPQ}}]} \|\mathbf{y}^{(m)} - \mathbf{d}_k^{(m)}\|_2, \quad \hat{\mathbf{y}}^{(m)} := \mathbf{d}_{d^{(m)}(\mathbf{z})}^{(m)}. \tag{16}$$

The OPQ reconstruction is $\hat{\mathbf{z}}_{\text{OPQ}} := R\hat{\mathbf{y}}$, where $\hat{\mathbf{y}} := [\hat{\mathbf{y}}^{(1)}; \ldots; \hat{\mathbf{y}}^{(M)}]$.

**Definition B.3** (RQ-OPQ Quantization). The RQ-OPQ quantization applies OPQ to the final RQ-Kmeans residual:

$$\hat{\mathbf{x}}_{\text{RQ-OPQ}} := \hat{\mathbf{x}}_{\text{RQ}} + \hat{\mathbf{z}}_{\text{OPQ}}, \quad \text{where } \mathbf{z} := r^{(L)}(\mathbf{x}). \tag{17}$$

The complete Semantic ID (SID) is the tuple:

$$\text{SID}_{\text{RQ-OPQ}}(\mathbf{x}) := \left( q^{(1)}(\mathbf{x}), \ldots, q^{(L)}(\mathbf{x}), d^{(1)}(r^{(L)}(\mathbf{x})), \ldots, d^{(M)}(r^{(L)}(\mathbf{x})) \right). \tag{18}$$

For notational convenience, we define the RQ-only SID as:

$$\text{SID}_{\text{RQ}}(\mathbf{x}) := \left( q^{(1)}(\mathbf{x}), \ldots, q^{(L)}(\mathbf{x}) \right). \tag{19}$$

### B.3.2. ERROR DECOMPOSITION ANALYSIS

**Theorem B.4** (Error Decomposition). *Let $\mathbf{x} \in \mathbb{R}^d$ be an item embedding. The reconstruction errors satisfy:*

1. *The RQ-Kmeans reconstruction error equals the final residual norm:*

$$\|\mathbf{x} - \hat{\mathbf{x}}_{RQ}\|_2^2 = \|r^{(L)}(\mathbf{x})\|_2^2. \tag{20}$$

2. *The RQ-OPQ reconstruction error equals the OPQ quantization error on the residual:*

$$\|\mathbf{x} - \hat{\mathbf{x}}_{RQ\text{-}OPQ}\|_2^2 = \|r^{(L)}(\mathbf{x}) - \hat{\mathbf{z}}_{OPQ}\|_2^2. \tag{21}$$

3. *The two encoding schemes exhibit **orthogonal decomposition**: RQ-Kmeans captures hierarchical coarse-to-fine structure through $\hat{\mathbf{x}}_{RQ}$, while OPQ independently refines the residual $r^{(L)}(\mathbf{x})$ without affecting the hierarchical encoding.*

*Proof.* **Part 1**: By Definition B.1, we have

$$\mathbf{x} - \hat{\mathbf{x}}_{\text{RQ}} = \mathbf{x} - \sum_{l=1}^{L} c_{q^{(l)}(\mathbf{x})}^{(l)} = r^{(L)}(\mathbf{x}), \tag{22}$$

where the last equality follows from the telescoping property of residuals. By induction on $l$:

$$r^{(l)}(\mathbf{x}) = \mathbf{x} - \sum_{j=1}^{l} c_{q^{(j)}(\mathbf{x})}^{(j)}, \quad \forall l \in [L]. \tag{23}$$

**Part 2**: By Definition B.3:

$$\begin{aligned} \mathbf{x} - \hat{\mathbf{x}}_{\text{RQ-OPQ}} &= \mathbf{x} - \hat{\mathbf{x}}_{\text{RQ}} - \hat{\mathbf{z}}_{\text{OPQ}} \\ &= r^{(L)}(\mathbf{x}) - \hat{\mathbf{z}}_{\text{OPQ}}. \end{aligned} \tag{24}$$

**Part 3**: The orthogonal decomposition follows directly from the construction: the OPQ codes $(d^{(1)}, \ldots, d^{(M)})$ are computed solely from $r^{(L)}(\mathbf{x})$, which is the information discarded by RQ-Kmeans. The two components encode complementary information without interference. $\square$

### B.3.3. INFORMATION COMPLEMENTARITY

We now establish that RQ-OPQ preserves strictly more information about the original embedding than RQ-Kmeans alone.

**Theorem B.5** (Information Complementarity). *Let $\mathbf{x} \sim \mathcal{P}$ be a random item embedding. Denote:*

- $Q := SID_{RQ}(\mathbf{x}) = (q^{(1)}(\mathbf{x}), \ldots, q^{(L)}(\mathbf{x}))$ *as the RQ codes,*

- $D := (d^{(1)}(r^{(L)}(\mathbf{x})), \ldots, d^{(M)}(r^{(L)}(\mathbf{x})))$ *as the OPQ codes.*

*Then:*

$$I(\mathbf{x}; SID_{RQ\text{-}OPQ}) \geq I(\mathbf{x}; SID_{RQ}), \tag{25}$$

*where $I(\cdot; \cdot)$ denotes mutual information. Equality holds if and only if $D$ is a deterministic function of $Q$, i.e., the residual $r^{(L)}(\mathbf{x})$ contains no additional information beyond the RQ codes.*

*Proof.* By definition, $\text{SID}_{\text{RQ-OPQ}} = (Q, D)$. Using the chain rule for mutual information:

$$\begin{aligned}
I(\mathbf{x}; \text{SID}_{\text{RQ-OPQ}}) &= I(\mathbf{x}; Q, D) \\
&= I(\mathbf{x}; Q) + I(\mathbf{x}; D \mid Q) \\
&= I(\mathbf{x}; \text{SID}_{\text{RQ}}) + I(\mathbf{x}; D \mid Q).
\end{aligned} \tag{26}$$

Since mutual information is non-negative, $I(\mathbf{x}; D \mid Q) \geq 0$, which establishes the inequality.

For the equality condition: $I(\mathbf{x}; D \mid Q) = 0$ if and only if $\mathbf{x} \perp D \mid Q$, meaning $D$ provides no additional information about $\mathbf{x}$ beyond what $Q$ already captures. This occurs precisely when the residual $r^{(L)}(\mathbf{x})$ is fully determined by $Q$, i.e., all items sharing the same RQ codes have identical residuals. $\square$

**Corollary B.6** (Strict Information Gain). *If the residual distribution $\mathcal{Q}_L := \{r^{(L)}(\mathbf{x}) : \mathbf{x} \sim \mathcal{P}\}$ has positive variance within at least one RQ cluster, then:*

$$I(\mathbf{x}; SID_{RQ\text{-}OPQ}) > I(\mathbf{x}; SID_{RQ}). \tag{27}$$

*Proof.* If there exists an RQ code $q = (q^{(1)}, \ldots, q^{(L)})$ such that multiple items map to $q$ but have distinct residuals, then OPQ can potentially assign different codes to these residuals, yielding $I(\mathbf{x}; D \mid Q) > 0$. $\square$

*Remark* B.7 (Practical Implications). In e-commerce search, items within the same category often share similar hierarchical features (captured by RQ-Kmeans) but differ in fine-grained attributes such as color variants, size options, or brand-specific features. The residual $r^{(L)}(\mathbf{x})$ encodes these distinctive characteristics. Theorem B.5 guarantees that OPQ captures this complementary information, enabling better discrimination of similar items.

### B.3.4. DISCRIMINATION CAPACITY ANALYSIS

We formalize the relationship between quantization schemes and their ability to distinguish different items.

**Definition B.8** (Discrimination Probability). For a quantization scheme with SID mapping $\phi : \mathbb{R}^d \to \mathcal{S}$, the discrimination probability on dataset $\mathcal{X}_N = \{\mathbf{x}_1, \ldots, \mathbf{x}_N\}$ is:

$$P_{\text{disc}}(\phi; \mathcal{X}_N) := \frac{1}{\binom{N}{2}} \sum_{i<j} \mathbf{1}[\phi(\mathbf{x}_i) \neq \phi(\mathbf{x}_j)]. \tag{28}$$

**Theorem B.9** (Discrimination Dominance). *For any dataset $\mathcal{X}_N$ with distinct items:*

$$P_{disc}(SID_{RQ\text{-}OPQ}; \mathcal{X}_N) \geq P_{disc}(SID_{RQ}; \mathcal{X}_N). \tag{29}$$

*Equality holds if and only if no two items share the same RQ codes but different OPQ codes.*

*Proof.* For any pair $(\mathbf{x}_i, \mathbf{x}_j)$ with $i \neq j$, consider two cases:

**Case 1**: $\text{SID}_{\text{RQ}}(\mathbf{x}_i) \neq \text{SID}_{\text{RQ}}(\mathbf{x}_j)$.

Since $\text{SID}_{\text{RQ-OPQ}} = (\text{SID}_{\text{RQ}}, D)$, we have:

$$\text{SID}_{\text{RQ-OPQ}}(\mathbf{x}_i) \neq \text{SID}_{\text{RQ-OPQ}}(\mathbf{x}_j). \tag{30}$$

**Case 2**: $\text{SID}_{\text{RQ}}(\mathbf{x}_i) = \text{SID}_{\text{RQ}}(\mathbf{x}_j)$.

In this case, RQ-Kmeans fails to distinguish the pair. However, if:

$$(d^{(1)}(r^{(L)}(\mathbf{x}_i)), \ldots, d^{(M)}(r^{(L)}(\mathbf{x}_i))) \neq (d^{(1)}(r^{(L)}(\mathbf{x}_j)), \ldots, d^{(M)}(r^{(L)}(\mathbf{x}_j))), \tag{31}$$

then RQ-OPQ successfully distinguishes the pair.

Combining both cases:

$$\mathbf{1}[\text{SID}_{\text{RQ-OPQ}}(\mathbf{x}_i) \neq \text{SID}_{\text{RQ-OPQ}}(\mathbf{x}_j)] \geq \mathbf{1}[\text{SID}_{\text{RQ}}(\mathbf{x}_i) \neq \text{SID}_{\text{RQ}}(\mathbf{x}_j)]. \tag{32}$$

Summing over all pairs yields the result. $\square$

**Corollary B.10** (ICR Improvement). *The Independent Coding Rate (ICR) satisfies:*

$$ICR_{RQ\text{-}OPQ}(\mathcal{X}_N) \geq ICR_{RQ}(\mathcal{X}_N), \tag{33}$$

*where $ICR(\mathcal{X}_N) := |\{SID(\mathbf{x}_i) : i \in [N]\}|/N$.*

*Proof.* ICR equals the fraction of items with unique SIDs. By Theorem B.9, any pair distinguished by RQ-Kmeans is also distinguished by RQ-OPQ, and RQ-OPQ may additionally distinguish pairs that RQ-Kmeans cannot. Therefore, the number of unique SIDs under RQ-OPQ is at least that under RQ-Kmeans. $\square$

*Remark* B.11 (Connection to Empirical Results). Corollary B.10 directly explains our experimental observations in Table 7: RQ-OPQ achieves 91.91% ICR compared to RQ-Kmeans' 68.08%, a relative improvement of 35.1%. This substantial gain validates the theoretical prediction that OPQ's residual encoding effectively distinguishes items that share hierarchical features.

### B.3.5. INFORMATION CAPACITY ANALYSIS

**Theorem B.12** (Information Capacity Upper Bound). *Let $H(\cdot)$ denote Shannon entropy and let $SID := SID_{RQ\text{-}OPQ}(\mathbf{x})$ for $\mathbf{x} \sim \mathcal{P}$. Then:*

$$H(SID) \leq \sum_{l=1}^{L} \log_2 K_l + M \log_2 K_{OPQ}, \tag{34}$$

*with equality achieved if and only if:*

1. *All code components $(q^{(1)}, \ldots, q^{(L)}, d^{(1)}, \ldots, d^{(M)})$ are mutually independent, and*

2. *Each component is uniformly distributed over its range.*

*Proof.* Write $\text{SID} = (Q, D)$ where $Q := (q^{(1)}, \ldots, q^{(L)})$ and $D := (d^{(1)}, \ldots, d^{(M)})$. By the chain rule for entropy:

$$H(\text{SID}) = H(Q, D) \leq H(Q) + H(D) \leq \sum_{l=1}^{L} H(q^{(l)}) + \sum_{m=1}^{M} H(d^{(m)}), \tag{35}$$

where the first inequality follows from subadditivity (with equality iff $Q \perp D$), and the second from repeated application of the chain rule (with equality iff all components are mutually independent).

For discrete random variables taking values in a finite set $\mathcal{A}$, $H(X) \leq \log_2 |\mathcal{A}|$ with equality iff $X$ is uniformly distributed. Applying this to each component: $H(q^{(l)}) \leq \log_2 K_l$ and $H(d^{(m)}) \leq \log_2 K_{\text{OPQ}}$. Combining yields the result. $\square$

**Proposition B.13** (CUR-ICR Relationship). *For dataset $\mathcal{X}_N$, the Codebook Utilization Rate at layer $l$ is:*

$$CUR^{(l)}(\mathcal{X}_N) := \frac{|\{k \in [K_l] : \exists\, i \in [N]\ s.t.\ q^{(l)}(\mathbf{x}_i) = k\}|}{K_l}. \tag{36}$$

*Then:*

$$ICR(\mathcal{X}_N) \leq \min\left(1, \frac{\prod_{l=1}^{L}\left(K_l \cdot CUR^{(l)}(\mathcal{X}_N)\right) \cdot K_{OPQ}^M}{N}\right). \tag{37}$$

*Consequently, maximizing CUR at each layer is necessary (but not sufficient) for achieving high ICR.*

*Proof.* Let $\mathcal{U}^{(l)} := \{k \in [K_l] : \exists\, i \text{ s.t. } q^{(l)}(\mathbf{x}_i) = k\}$ be the set of utilized codewords at layer $l$. By definition, $|\mathcal{U}^{(l)}| = K_l \cdot \mathrm{CUR}^{(l)}$.

The number of distinct SIDs is bounded by the product of utilized codes:

$$|\{\mathrm{SID}(\mathbf{x}_i) : i \in [N]\}| \leq \prod_{l=1}^{L} |\mathcal{U}^{(l)}| \cdot \prod_{m=1}^{M} |\mathcal{D}^{(m)}| = \prod_{l=1}^{L} \left( K_l \cdot \mathrm{CUR}^{(l)} \right) \cdot K_{\mathrm{OPQ}}^{M}. \tag{38}$$

Dividing by $N$ and noting $\mathrm{ICR} \leq 1$ yields the bound. $\qquad\square$

### B.3.6. ADVANTAGE OF OPQ ROTATION

**Definition B.14** (Residual Covariance). Let $\Sigma_L := \mathbb{E}_{\mathbf{x} \sim \mathcal{P}}[r^{(L)}(\mathbf{x}) r^{(L)}(\mathbf{x})^\top] \in \mathbb{R}^{d \times d}$ denote the covariance matrix of the final RQ-Kmeans residual (assuming zero mean).

**Definition B.15** (Rotated Sub-covariance). For rotation matrix $R$ and subspace $m \in [M]$, let $P_m \in \mathbb{R}^{(d/M) \times d}$ be the projection matrix selecting coordinates $\{(m-1)d/M + 1, \ldots, md/M\}$. The $m$-th sub-covariance after rotation is:

$$\Sigma_L^{(m)}(R) := P_m R^\top \Sigma_L R P_m^\top \in \mathbb{R}^{(d/M) \times (d/M)}. \tag{39}$$

**Theorem B.16** (OPQ Rotation Properties). *The following statements characterize the role of rotation in OPQ:*

1. *(**Variance Invariance**) The total variance is invariant under rotation:*

$$\sum_{m=1}^{M} Tr\left( \Sigma_L^{(m)}(R) \right) = Tr(\Sigma_L), \quad \forall R \in O(d), \tag{40}$$

   *where $O(d) := \{R \in \mathbb{R}^{d \times d} : R^\top R = I_d\}$ is the orthogonal group.*

2. *(**Decorrelation Benefit**) Product quantization assumes independence across subspaces. The optimal rotation $R^*$ improves quantization efficiency by minimizing statistical dependence between subspaces, i.e., by reducing the off-diagonal blocks of $R^\top \Sigma_L R$.*

*Proof.* **Part 1**: Using the cyclic property of trace and the fact that $\{P_m\}$ form a partition of coordinates:

$$\begin{aligned}
\sum_{m=1}^{M} \mathrm{Tr}\left( \Sigma_L^{(m)}(R) \right) &= \sum_{m=1}^{M} \mathrm{Tr}\left( P_m R^\top \Sigma_L R P_m^\top \right) \\
&= \sum_{m=1}^{M} \mathrm{Tr}\left( P_m^\top P_m R^\top \Sigma_L R \right) \\
&= \mathrm{Tr}\left( \sum_{m=1}^{M} P_m^\top P_m \cdot R^\top \Sigma_L R \right) \\
&= \mathrm{Tr}(I_d \cdot R^\top \Sigma_L R) = \mathrm{Tr}(\Sigma_L),
\end{aligned} \tag{41}$$

where we used $\sum_m P_m^\top P_m = I_d$ and the invariance of trace under orthogonal similarity transformations.

**Part 2**: Product quantization treats subspaces independently. When the rotated residual $R^\top r^{(L)}$ has correlated subspaces (non-zero off-diagonal blocks in $R^\top \Sigma_L R$), the independence assumption is violated, leading to suboptimal quantization. The optimal rotation $R^*$ seeks to block-diagonalize $R^\top \Sigma_L R$, making subspaces as independent as possible. $\qquad\square$

*Remark* B.17 (Clarification on Error Reduction Mechanism). Theorem B.16 Part 1 shows that rotation does not reduce total variance—this quantity is invariant. The benefit of OPQ lies entirely in making the independence assumption of product quantization more accurate. When subspaces are decorrelated, independent quantization achieves performance closer to optimal joint quantization.

B.3.7. SUMMARY OF THEORETICAL GUARANTEES

The theoretical analysis establishes the following key properties of RQ-OPQ:

1. **Information Complementarity** (Theorem B.5): RQ-OPQ provably captures at least as much information about the original embedding as RQ-Kmeans, with strict improvement when residuals contain non-trivial information within RQ clusters.

2. **Discrimination Dominance** (Theorem B.9): RQ-OPQ can distinguish at least as many item pairs as RQ-Kmeans, directly implying improved ICR. This explains the empirical observation in Table 7 (91.91% vs. 68.08% ICR).

3. **Information Capacity** (Theorem B.12): The semantic ID can encode up to $\sum_{l=1}^{L} \log_2 K_l + M \log_2 K_{\text{OPQ}}$ bits of information, with the OPQ component contributing additional $M \log_2 K_{\text{OPQ}}$ bits beyond RQ-Kmeans.

4. **Utilization-Uniqueness Trade-off** (Proposition B.13): High codebook utilization (CUR) is necessary for high independent coding rate (ICR), establishing a quantitative relationship between codebook design and item distinguishability.

5. **Rotation Benefit** (Theorem B.16): OPQ's learnable rotation improves quantization efficiency by decorrelating subspaces, enabling independent quantization to better approximate optimal joint quantization.

These results provide theoretical justification for the KHQE module's design: RQ-Kmeans captures hierarchical semantic structure shared across similar items, while OPQ encodes complementary fine-grained features that distinguish items within the same cluster. The combination ensures both semantic coherence (items with similar meanings share RQ prefixes) and discrimination capacity (distinct items receive unique SIDs), which are essential for effective e-commerce search.

# C. Implementation Details

We adopt Bart-B (Lewis et al., 2020) as the base pre-trained model for the testing and online deployment, as it is an efficient model with optimized architectural acceleration, and has been online applied in many scenarios in Kuaishou. Due to commercial confidentiality, we do not disclose the total parameters of the online model here, but it is at least 100 times larger than Bart. The beam search size is set to 512 here to strike a balance between generation quality and latency. The maximum window length is set to n=5. The batch size for SFT and DPO is set to 512 and 128, respectively, with the latter being smaller because the list-wise DPO training takes more samples as inputs. For RQ-OPQ, the number of codebook layers $C = 5$ (3 layers for RQ-Kmeans, and 2 layers for residual OPQ). The codebook size $W$ of each layer is (4096,1024,512—256,256). Some of the hyperparameters will be discussed in the following ablation study. The multi-stage supervised training is conducted every week, RL with the reward system is conducted daily, and the hybrid preference alignment with user interaction data is updated as close to the stream as possible. Actually, RL with a reward system can also be trained every week, as we found it does not bring significant performance gains, except during the global shopping festivals (e.g., 11.11 and 6.18).

## D. Ablation study of different OPQ tokenizations

We examined the impact of different hierarchical quantization encodings on items in Figure 6. As shown in Table 8, we computed two metrics with the top 10 items for quick validation. RQ-OPQ (2/256) is the basic configuration, and RQ-OPQ (4/256) means the residual embedding is tokenized by OPQ (256-256-256-256). RQ-OPQ (4*2/256) means all embeddings (the cluster of three layers and the residual one) are tokenized with OPQ (2/256), then (4*4/256) indicates further quantization. We found that the basic RQ-OPQ (2/256) achieved the highest performance. (4/256) perform weakly with increased sequence length and decoding complexity. The other two configurations were almost entirely ineffective, which is similar to the balanced k-means operation on full layers in § 3.1, as the hierarchical features were not distinctly represented, leading to many items being aggregated under the same SID.

*Table 8.* Ablation study of different OPQ tokenizations.

| Method | order (30k) | | click (30k) | |
| --- | --- | --- | --- | --- |
| | **HR@10** | **MRR@10** | **HR@10** | **MRR@10** |
| RQ-OPQ (2/256) | 28.42% | 14.15% | 33.69% | 11.94% |
| *-OPQ (4/256) | -2.36% | -1.77% | -2.52% | -1.56% |
| *-OPQ (4*2/256) | -10.20% | -5.57% | -11.77% | -3.84% |
| *-OPQ (4*4/256) | -24.18% | -11.83% | -27.11% | -9.61% |

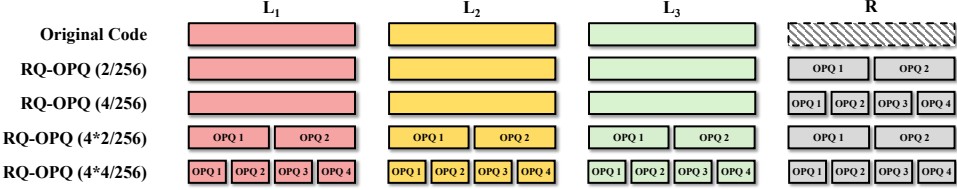

*Figure 6.* The different hierarchical quantization encodings of items.

## E. Manual evaluation results for online experience

Last but not least, to ascertain the actual impacts on the online search experience, we conducted additional manual evaluations. We randomly selected 200 queries and extracted 3,200 query-item pairs from identical exposure positions, ensuring all other variables remained constant. We set three metrics as 1) page good rate - an evaluation indicator for the overall user experience, 2) item quality - Check whether the displayed products are counterfeit, have mismatched images and text, or have abnormal prices, and 3) query-item relevance - we engaged experts to rate each pair as "Good" (both subject and core keywords match), "Fair" (only subject matches), or "Bad" (subjects differ). The outcomes of these assessments are presented in Table 9. We can see that OneSearch[2] achieves substantial increases in page good rate by 1.03%, item quality by 2.12%, and query item relevance by 1.87%. The deployment of RQ-OPQ further enhances the relevance of model generation.

*Table 9.* Manual evaluation results for online experience.

| Metric | Page Good Rate | Item Quality | Q-I Relevance |
|---|---|---|---|
| OneSearch[1] | 0.84% | 1.69% | 1.40% |
| OneSearch[2] | 1.03% | 2.12% | 1.87% |

# F. Further Studies

**What are the main aspects of the online gains for the OneSearch?** In our analysis, we focused on the dimensions of industry and query popularity. As illustrated in Figure 7, we calculated the CTR relative gains across the top 30 industries. Remarkably, 28 out of 30 industries experienced increases, with an average gain of 2.49%. These results were statistically significant, with P-values below 0.05. Although two industries showed negative effects, these were not statistically significant. Overall, the unified modeling optimization demonstrates substantial potential in addressing the inconsistent objectives of multi-stage processes in MCA systems, benefiting nearly all industries.

As for the query popularity dimension, we divided all prefixes into three categories: top (PV number daily larger than 1,000), middle (larger than 100 and less than 1,000), and long-tail (less than 100). The item CTR relative gains for each were listed in Table 10. Queries of all categories are enhanced with the OneSearch models. These results indicate that the rich semantic and interactive representations induced by keyword-enhanced hierarchical quantization encoding, multi-view behavior sequence, and the preference aware reward system can greatly improve the recognition of e-commerce search for queries of all popularity.

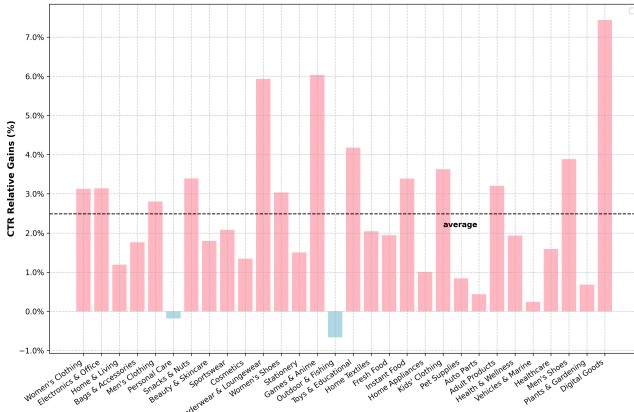

*Figure 7.* The online CTR relative gains for top 30 industries.

*Table 10.* Online CTR gains for three query popularity.

| Method | Top | Middle | Long-tail |
|---|---|---|---|
| OneSearch[2] | +1.25% | +2.27% | +1.33% |

**Does OneSearch have stronger reasoning capabilities?** In traditional e-commerce search scenarios, ranking models often involve thousands of features, and the combination of them can obscure some key attributes. Additionally, the structure of common ranking model typically consists of a simple stack of shallow neural networks, resulting in minimal reasoning capabilities. OneSearch, on the other hand, leverages users' long- and short-term sequential information to identify their potential interests and enhances the inference of user search intent through the attention mechanism of transformer structures. For instance, a female user who previously searched for "couple sneakers" and "Valentine's Day gifts" is likely seeking a pair of rings for both her partner and herself when searching for "silver ring." We observed in real logs that only OneSearch presented the relevant product, which was ultimately purchased by the user.

**How does OneSearch perform for cold-start users and items?** We conducted tests to evaluate the model's performance in cold-start scenarios. Here, we define cold items as those published within the last seven days with no interaction behavior, and cold users as those who have not used the Kuaishou app in the past 90 days. The specific comparison results are demonstrated in Table 11. Compared to the onlineMCA, we found that OneSearch's performance for cold-start items and users has improved by 3.31% and 2.50%, respectively. Both of them are greater than the metrics for warm ones. These results show that OneSearch can handle the cold-start issue well.

**What potential optimization opportunities will OneSearch explore in the future?** The addition of OPQ-based tokenization can even quickly process new hotwords. We constructed a new keyword offline and added it to the textual descriptions

*Table 11.* Online CTR gains for cold-start items and users.

| Object | Warm | Cold | Average |
|--------|--------|--------|---------|
| Item | +2.34% | +3.31% | +2.52% |
| User | +1.11% | +2.50% | +2.41% |

of some items. Without reconstructing a new codebook, OneSearch was still able to generate SIDs for these items during inference. This finding further motivates us to consider online real-time encoding. We will explore in future research, aiming to achieve unified encoding and inference using a single generative model, thereby reducing the gap between scheduled encoding and streaming training phrase. Additionally, aligning user preferences through more robust reinforcement learning and incorporating multi-modal features (such as images and videos) for items can further enhance the reasoning capabilities.

