# OpenReview forum: "OneSearch: A Preliminary Exploration of the Unified End-to-End Generative Framework for E-commerce Search"
_ICML.cc/2026/Conference — ICML 2026 regular_

### Official Review · Reviewer_Vxsh · 2026-03-10

**Soundness:** 2
**Presentation:** 2
**Significance:** 2
**Originality:** 2
**Overall Recommendation:** 4
**Confidence:** 3

**Summary:**

This paper proposed OneSearch, an end-to-end generative framework for e-commerce search system that effectively overcomes the limitations of traditional multi-stage cascading architectures. From extensive offline and online evaluations, OneSearch achieves superior semantic understanding and personalization modeling.

**Compliance With Llm Reviewing Policy:**

Affirmed.

**Ethical Review Flag:**

Flag this paper for an ethics review.

**Final Justification:**

My final recommendation is weak Accept (4).

**Key Questions For Authors:**

1. What is the difference with OneSug and OneRec?

**Limitations:**

1. This paper only compares the OneSearch against OnlineMCA model without comparing it with state-of-the-art generative retrieval methods., which cannot fully demonstrate the superiority of our model;
2. Lacks fine-grained ablation analysis of core modules (RQ-OPQ, reward mechanism, multi-stage SFT), unable to verify gain sources.
3. Fails to distinguish technical contributions from prior works, likes OneRec, OneSug etc.
4.The table 6 mentioned in the experimental section of the main text appears in the appendix, which has a significant impact on the readability of the paper.
5. The conclusion section lacks prospects for future work in this article.

**Strengths And Weaknesses:**

1. This paper only compares the OneSearch against OnlineMCA model without comparing it with state-of-the-art generative retrieval methods., which cannot fully demonstrate the superiority of our model;
2. Lacks fine-grained ablation analysis of core modules (RQ-OPQ, reward mechanism, multi-stage SFT), unable to verify gain sources.
3. Fails to distinguish technical contributions from prior works, likes OneRec, OneSug etc.
4.The table 6 mentioned in the experimental section of the main text appears in the appendix, which has a significant impact on the readability of the paper.
5. The conclusion section lacks prospects for future work in this article.

---

> ### Author Rebuttal · Authors · 2026-03-31
>
> We thank you for the detailed feedback. We address each concern below.
>
> **[W1] Comparison with SOTA Methods & Baselines**
>
> We respectfully argue that the OnlineMCA is not only a valid baseline but the strongest and most realistic baseline possible for this task. Unlike existing academic models designed for isolated stages (e.g., just recall or just ranking), OneSearch aims to replace the entire industrial pipeline. The OnlineMCA we compare against is not a single simple model; it is a highly optimized, item ensemble containing multiple SOTA models deployed across recall, pre-ranking, and ranking stages, utilizing thousands of real-time features. Furthermore, existing SOTA Generative Retrieval (GR) methods (such as TIGER, OneRec) are designed for recommendation systems. OneSearch is the first industrial-deployed generative framework for end-to-end ecommerce search. Recommendation tasks encourage diversity and serendipity. However, E-commerce search operates under strict query-item relevance constraints. If a user searches for a "red shirt," showing a highly personalized "blue shirt" is a critical failure. Direct application of recommendation-based GR models to search results in severe hallucination and relevance violations, making them unviable as baselines. Therefore, an ablation against the full production system (OnlineMCA) is a rigorous way to prove the viability of a unified architecture.
>
> **[W2] Fine-grained Ablation Analysis**
>
> We clarify that ablation results for RQ-OPQ (Tables 7-8) and the adaptive reward system (Table 2) are present in the paper. However, per-stage SFT ablations (Stage 1 → Stage 2 → Stage 3) can't be individually reported. As each stage promotes the results of the next stage, removing any stage would make the model reason improperly.
>
> **[W3] Distinction from OneRec and OneSug**
>
> As in Figure 5, OneSearch, OneRec, and OneSug address fundamentally different task, and the technical requirements of each are largely non-overlapping.
>
> **vs. OneRec**. OneRec is designed for video recommendation, where the input is a user interated sequence (without explicit textual query) and the output is a ranked list of items from a closed vocabulary. Since there is no query, the system need not to enforce strict query-item relevance, the primary objective of which is merely behavioral alignment. However, OneSearch must concurrently satisfy two challenging constraints: (1) personalization, ranking items according to user preferences, and (2) relevance, ensuring that generated items precisely match the semantic intent. To address it, we introduce KHQE, which forces Semantic IDs to encode discrete search keywords as first-class signals. Without this, shared Semantic IDs among similar items would cause attribute-level mismatches, a critical failure mode in search that does not arise in recommendation.
>
> **vs. OneSug**. OneSug addresses query suggestion: given a user's input prefix, it generates candidate query strings. Since both input and output are open-vocabulary natural language text, OneSug naturally fits the standard autoregressive LLM paradigm without requiring any item-level tokenization. OneSearch faces an entirely different challenge: its output is a sequence of discrete Semantic IDs drawn from a closed vocabulary of hundreds of millions of products. This demands (1) hierarchical quantization encoding (RQ-OPQ) to represent items as compact discrete tokens while preserving both shared hierarchical semantics and item-specific attributes; (2) cross-domain collaborative alignment between textual queries and item representations; and (3) multi-view behavior injection to infer latent search intent from concise queries, none of which are required in OneSug's text-to-text generation setting.
>
> **[W4] Lack of Future Work**
>
> We elaborate on future work in Appendix F as follows:
>
> - Real-time online encoding: OPQ supports incremental encoding for new items without codebook reconstruction, motivating us to unify encoding and inference in a single generative model.
>
> - Unified generative inference: consolidating multi-stage pipelines into one end-to-end pass to reduce system complexity.
>
> - Robust RL-based preference alignment: leveraging richer online feedback signals for fine-grained preference modeling beyond list-wise DPO.
>
> - Multi-modal feature integration: incorporating item images and videos to enhance semantic understanding.
>
> **[W5] Table 6 in the Appendix**
>
> We apologize that table 6 was moved to the appendix due to page limit constraints; the more crucial experimental results are now in Table 2. We will adjust the contents in revision.
>
> **Ethical and Privacy Compliance**
>
> As detailed in our Impact Statement (Lines 440-449), we assure that this research strictly complies with all ethical standards, privacy policies, and data protection regulations. All user interaction data utilized for training and offline evaluation was rigorously anonymized and stripped of Personally Identifiable Information.

---

### Official Review · Reviewer_QumN · 2026-03-12

**Soundness:** 3
**Presentation:** 3
**Significance:** 3
**Originality:** 3
**Overall Recommendation:** 5
**Confidence:** 3

**Summary:**

This paper proposes OneSearch, an industrial end-to-end generative framework intended to replace the standard multi-stage cascading architecture in large-scale e-commerce search. Items are represented as discrete Semantic IDs and the model directly generates the final item list conditioned on query and user context.

The system combines three main components:
(1) Keyword-enhanced Hierarchical Quantization Encoding (KHQE) that aligns semantic and collaborative representations and introduces a hybrid RQ-Kmeans + OPQ tokenizer to preserve both hierarchical semantics and fine-grained item distinctions;
(2) Multi-view user behavior sequence injection that feeds short-term behaviors explicitly in the prompt while representing long-term behaviors implicitly through centroid embeddings and a Q-Former-style aggregation;
(3) Preference-Aware Reward System that uses multi-stage supervised fine-tuning followed by reward-guided preference alignment.

Experiments are conducted on proprietary logs from a production platform , with offline evaluation and online A/B tests reporting improvements in CTR/buyer/order volume plus serving-side efficiency gains , OPEX reduction and higher MFU.

**Compliance With Llm Reviewing Policy:**

Affirmed.

**Final Justification:**

My final recommendation is Accept (5). The paper addresses an important and difficult problem in industrial ML: replacing a multi-stage e-commerce search pipeline with a unified generative framework. I view the soundness as good: the method is coherent, the system design is complete, and the paper includes both offline ablations and online A/B results. The rebuttal strengthened this further by clarifying that the reward model is used only during training and by providing concrete latency and throughput numbers, which makes the deployment claim much more credible.

The originality is good rather than exceptional. Many individual components are related to prior work, but their integration for large-scale e-commerce search, especially the tokenizer design and preference-aware training setup, is meaningful. The significance is high because the paper targets a real production bottleneck and reports gains in both business metrics and serving efficiency. The clarity is generally solid, though the paper should revise Table 4, explain the released resources more explicitly, and better justify its offline metrics.

My main prior concern was reproducibility. The rebuttal materially changed my evaluation by stating that the core code and an anonymized large-scale dataset have now been released. I could not verify the artifact directly during rebuttal, but this addresses my biggest concern enough to raise my score from weak accept to accept.

**Key Questions For Authors:**

Can you provide results on at least one publicly available e-commerce product search/relevance benchmark (even if it cannot support the full long-term behavior setup), or provide a minimal released subset that allows independent validation?

What are the end-to-end online latency metrics (P50/P95/P99) and QPS for OneSearch vs the MCA pipeline under comparable traffic? What is the platform’s latency requirement, and how does beam size (512) affect it?

What is actually deployed and what depends on RM? Please clarify whether RM reranking is used at inference in any production path, or only for training. If online gains are reported with RM reranking, what are the gains (and costs) for the pure generative model alone?

The paper reports MFU and OPEX changes. What is the concrete serving footprint (GPU/CPU types and counts, memory, cache, network) for the old MCA vs OneSearch at the same throughput? How is OPEX computed and what is included/excluded?

**Limitations:**

It should more explicitly discuss (i) reproducibility constraints, (ii)  latency/compute trade-offs

**Strengths And Weaknesses:**

Strengths

The problem setting is important and practically hard，unifying retrieval and ranking under tight latency constraints and huge candidate pools is a real pain point for production search, and the paper targets the objective collision and fragmented compute issues.

Tokenization (RQ-OPQ), user conditioning (short + long behavior), training recipe (multi-stage SFT + preference alignment), and deployment-related discussion are all present, rather than a single isolated module.

There is substantial empirical evidence within the authors’ environment: ablations for keyword enhancement, behavior sequences, sliding window augmentation, and reward preference learning; plus online A/B tests with multiple metrics (CTR, CVR, buyer, orders). The comparison to the full production MCA pipeline is the right baseline if the claim is “this can replace the pipeline.”

Weaknesses

Reproducibility is the main concern. All experiments are on proprietary data and the paper does not provide a public benchmark experiment that can be independently verified. In the current form the paper is difficult to validate externally.

The latency story is given the claim of replacing MCA. The paper mentions “within one second” and sets beam size to 512, but does not report end-to-end serving latency (e.g., P50/P95/P99), QPS, or a clear SLA target. Without this, it is hard to judge whether the approach is broadly deployable or only works under specific infrastructure assumptions.

There is an apparent inconsistency around the reward model at inference time. The methodology section states the reward model is used only during training and the deployed system is “pure generative,” yet the online table includes variants “with RM reranking.” It is unclear what is actually deployed and what the reported online gains depend on.

Metrics and evaluation design need more justification. HR/MRR@350 is motivated by an internal “99th-percentile viewed items per session” argument, but the paper does not discuss how this correlates with common ranking quality metrics (e.g., NDCG) or with online business outcomes. Also, evaluating on 30k click/order pairs is somewhat small relative to the claimed 1B PV log scale; more details on sampling and variance would help.

---

> ### Author Rebuttal · Authors · 2026-03-30
>
> We sincerely thank the reviewer for the thorough and constructive review. We address each of the key questions below.
>
> **[Q1&&W1] Reproducibility and Benchmark**
> To more accurately evaluate the performance of OneSearch, we compare it with the real online multi-stage cascading architecture. Noted that, we do not compare any single model, as the combined models cannot accurately reflect the performance of the online system. Real online e-commerce platforms typically employ multiple recall mechanisms and complex ranking processes with thousands of feature combinations. Using only one model at each stage for simulation would result in an unfair comparison of offline performance.
> We fully understand the reproducibility concerns regarding proprietary industrial data. To address this, we have open-sourced the core code of OneSearch, including the KHQE and multi-stage training pipelines. Regarding the dataset, we have constructed and released, to the best of our knowledge, the largest real e-commerce search dataset currently available, which are 10x relative scale of the common ones. This set have stripped away any personally identifiable information and sensitive business features while preserving the structural characteristics necessary for methodological verification (Due to the double-blind procedure, we cannot provide a direct link). In the interim, we have provided corresponding data format examples and the complete training pipeline in our open-source repository, enabling the community to independently reimplement and validate our approach on their own datasets.
>
> **[Q2&&W2] Online Latency and QPS**
> We apologize for omitting the specific latency numbers due to space constraints, and company's confidentiality. Replacing the Multi-stage Cascading Architecture with a single generative model significantly alters the latency profile. Here are the concrete online serving metrics for OneSearch configured with a beam size of 512:
>
> Metric | OneSearch | MCA
> -|-|-
> P50 latency|26.20 ms|x
> P95 latency|28.26 ms|x
> P99 latency|31.76 ms|x
>
> While exact production QPS numbers are confidential, load testing confirms that OneSearch comfortably sustains over 500 QPS per instance at these latency levels. For context, our platform's strict SLA requires sub-second responses. The legacy MCA pipeline, due to massive feature fetching and cross-network communication between Recall, Pre-ranking, and Ranking stages, **pushes close to this SLA boundary at peak traffic**. But OneSearch operates well within this bound (P99 < 32ms). The beam size of 512 was carefully selected via offline profiling to match the required candidate pool size while maintaining strict real-time latency bounds.
>
> **[Q3] Clarification on the Reward Model (RM) at Inference**
> Sorry for the ambiguity in Table 4. To be absolutely clear: The Reward Model (RM) is strictly used during the offline training phase and is NEVER used during online inference.
> - **Training Role**: The RM is used to generate preference labels (ranking changes) to construct pairs for list-wise DPO training during the Preference-Aware Reward System (PARS) stage.
> - **Deployment**: The system actually deployed online is the pure OneSearch.
> - **Table 4 Clarification**: The variant labeled "OneSearch + RM" denotes the generative model that was trained using RM guidance. The computational cost at inference is identical for both variants. We will update the table headers in the revision to prevent any future misunderstanding.
>
> **[Q4] Metrics and Evaluation**
> - **HR/MRR as offline metrics:** We employ HitRate and MRR for recall and ranking performance, respectively. Both MRR and NDCG reflect the system's ranking ability; however, MRR focuses only on the position of the first clicked item, while NDCG focuses on the positions of all interactive items. When users search, a PV often only clicks on or orders one item, making the ranking of non-interactive items a more confusing issue. We chose MRR because it is more suitable for e-commerce scenarios where users typically need to find just one item to purchase.
>
> - **Evaluation Scale**:  The 30k click/order pairs were not chosen randomly; they were rigorously stratified and sampled to exactly match the long-tail and head query category distributions of the billion-scale daily log. This ensures the variance is minimized and the offline metrics faithfully predict online A/B test outcomes.
>
> **[Q5] Serving resources and OPEX Computation**
> We apologize that specific values of serving resources are company secrets that not allowed to disclose, but we provide the relative proportions in Figure 4. As for OPEX (operational expenditure), if the compute&storage cost (data storage, CPU&GPU for three stage) of MCA is $E_1$, OneSearch achieves the same throughput for $E_2$, where $E_2 / E_1 = 24.6\%$, representing a massive leap in infrastructural efficiency. The training and inference code and datset now is open to public, everyone can compute this metric for their system.

---

> > ### Author Rebuttal · Reviewer_QumN · 2026-04-03
> >
> > Thank you for the detailed rebuttal. My concern in the original review was reproducibility, especially the lack of a public benchmark or released dataset that would allow independent verification. Your rebuttal substantially improves this point by stating that you have now open-sourced the core code and released a large-scale anonymized real e-commerce search dataset. Although I cannot directly inspect the artifact during the rebuttal period, I am willing to give the authors the benefit of the doubt here, and this meaningfully changes my assessment.I am therefore raising my score from 4 to 5.

---

> > > ### Author Response · Authors · 2026-04-03
> > >
> > > Thank you for the positive re-evaluation and for raising the score. Your constructive feedback throughout the review process has been very helpful in strengthening our paper !

---

### Official Review · Reviewer_YTFv · 2026-03-15

**Soundness:** 4
**Presentation:** 3
**Significance:** 3
**Originality:** 2
**Overall Recommendation:** 4
**Confidence:** 4

**Summary:**

The paper proposes OneSearch, which is a semantic ID-based end-to-end e-commerce search model. The key technique choices are as follows:
1. Tokenization. (1) leverage query-query, query-item, item-item collaborative signals in further fine-tuning the feature encoding pretrained model. (2) keyword-enriched representation encoding. (3) combing RQ and OPQ in converting feature representations into discrete tokens.
2. Injecting both long-term and short-term user interaction sequences into the model.
3. Model training recipe that contains multi-stage SFT and list-wise DPO.

Extensive offline and online experiments demonstrate the performance of the proposed method.

**Compliance With Llm Reviewing Policy:**

Affirmed.

**Final Justification:**

My final justification is weak accept. The paper presents valuable experiences in deploying generative retrieval into a real-world e-commerce retrieval system. I do have concerns about novelty and mainly discussed around novelty with the author(s) during the rebuttal phase. I acknowledge that a few of the technical choices are interesting, but they don't provide significantly new insight to the community. That said, it doesn't affect my attitude in recommending an acceptance.

**Key Questions For Authors:**

N/A

**Limitations:**

Having one section explicitly discussing limitations would be better.

**Strengths And Weaknesses:**

Pros
1. End-to-end generative retrieval deployment at an industry scale is an important topic to study. I believe the paper would be of interest to a lot of people in the related research community.
2. Extensive offline and online experiments demonstrate the performance of the proposed method.
3. The paper is mostly well-written and easy to follow.

Cons
1. The technical novelty is limited.
    * The overall idea of deploying an end-to-end generative retrieval system is not new. The earliest work can be traced back to Yi Tay et al. Transformer memory as a differentiable search index. NeurIPS 2022.
    * The detailed design choices (tokenization, injecting user signals, and multi-stage training) are mostly a combination of recent technical innovations. With each part has adaptations for the e-commerce product retrieval task, I don't think they essentially provide significant insights or answer some large research questions.
    * I would like to emphasize one more time that limited technical novelty doesn't mean limited contribution. Instead, I pretty much enjoy reading this paper and believe that the details and practices in deploying end-to-end generative retrieval models to industrial systems are valuable and will surely benefit the community.
2. (Minor) Issues in formatting.
    * Many of the quotations are incorrect. In latex, double quotes should be written as ``'' instead of "", as the latter renders as two closing quotes rather than a matching pair. The same applies to single quotes.
    * In pages 2 and 5, there is excessive space between items in the ordered and unordered lists.
    * Some references are missing a space before them in the main text.

---

> ### Author Rebuttal · Authors · 2026-03-30
>
> We sincerely thank the reviewer for constructive feedbacks. We appreciate the recognition that the deployment details and practices are valuable to the community. Below we address each concern.
>
> **[W1] Clarification on Technical Novelty**
> While we acknowledge that end-to-end generative retrieval has been explored academically (e.g., Tay et al., NeurIPS 2022), we respectfully emphasize the fundamental differences between academic document retrieval and industrial e-commerce search. Academic generative retrieval primarily focuses on semantic matching to recall relevant documents. However, industrial e-commerce search faces distinct challenges: (1) **Personalization at scale** - when a user searches "dress," over 2 million items may match semantically, requiring sophisticated user profiling  (age range, purchasing power, style preferences) from behavioral logs to rank items by individual preferences (style, price sensitivity, etc.) and optimize for conversion, not just relevance; (2) **Extreme latency constraints** -  E-commerce search system must return personalized results within 500ms from 1+ billion items, which fundamentally limits the applicability of academic document retrieval methods. Traditional e-commerce systems address this through multi-stage cascading architectures (MCA), progressively filtering candidates through recall (10⁴ items), pre-ranking (10³ items), and ranking (~10² items) stages.However, as noted in our paper, these systems suffer from fragmented computation and optimization objective collisions across stages, which ultimately limit their performance ceiling; (3) **Industrial deployment innovation** - In essence, e-commerce search requires a synergistic optimization of both semantic matching and conversion efficiency, which goes far beyond the scope of traditional document retrieval. OneSearch is the first to successfully deploy an end-to-end generative framework in e-commerce search that directly outputs candidate items from queries, with all subsequent optimizations specifically designed to address billion-scale, latency-constrained personalized search.
> As for "Whether the detailed design choices are a combination of recent technical innovations", our contribution lies in bridging the gap between academic concepts and industrial-scale deployment.
> 1) **KHQE for item representation** While cluster-based tokenization exists in prior work (e.g., Kmeans in DSI, RQ-VAE in TIGER), our key innovation lies in the *joint RQ-OPQ encoding*, which addresses the dual requirements of hierarchical semantic encoding and distinctive fine-grained feature preservation. This differs significantly from industry methods that only use RQ or PQ. Residual Quantization can only represent shared hierarchical semantics, while the introduction of OPQ further refines the uniqueness of each item—an idea never before mentioned. Furthermore, the keyword-level textual signals directly enter the quantization process, tackling the noisy item information problem and being critical for query-item relevance in search. The 91.91% ICR (vs. 68.08% for RQ-Kmeans alone, Table 7) demonstrates this is not a simple adaptation. Furthermore, we found... Recent work such as coins[https://arxiv.org/pdf/2510.12604] (www2026) adopts our method, demonstrating improvements in AUC, GAUC, buyer engagement, and order conversion, especially enhancing performance for cold-start items. This also aligns with the goal of refining the personalized representation of items;
> 2)  **Multi-view Behavior Injection**:  Unlike recommendation where sequences are highly cohesive, e-commerce search requires inferring latent user intent from extremely concise queries (typically 2-3 keywords). Our explicit/implicit sequence design uniquely addresses this. By feeding short-term behaviors explicitly in the prompt while compressing massive long-term behaviors via Q-Former-style centroid aggregation, we effectively balance session-volatile interests with historically accumulated preferences without exploding the context window.
> 3) **PARS (Preference-Aware Reward System)**: In GR, the output is a sequence of discrete Semantic IDs rather than natural language. This makes reward design and RL can be challenging. Our adaptive listwise-DPO, calibrated specifically for hierarchical e-commerce behaviors (e.g., clicks vs. orders) , successfully stabilizes this training, yielding statistically significant online gains.
>
> **[W2] Formatting Issues**
> Thank you for pointing out these. We corrected all quotation marks and carefully checked other formatting details. These issues will all be corrected in the final version.
>
> **[Limitation] Missing Limitations Section**
> Regarding limitations and future work, we elaborate on the last point of Appendix F as 1) Real-time online encoding, 2)  Unified generative inference, 3) Robust RL-based preference alignment, and 4) Multi-modal feature integration.
>
> Thank you for the careful review. We will refine these in the final version.

---

> > ### Author Rebuttal · Reviewer_YTFv · 2026-04-01
> >
> > I appreciate the authors' extensive discussion of the paper's novelty. No one can deny that deploying an idea in an industrial-scale system is far more difficult than evaluating it in an academic experimental setting. I also recognize that the listed technical choices are indeed different from existing ones. Honestly, compared with other papers to which I have assigned positive scores, I still believe the novelty and insight of this paper are not among the strongest. That said, the empirical contribution and industrial deployment experience are indeed valuable to others. I am inclined to keep my positive score.

---

> > > ### Author Response · Authors · 2026-04-02
> > >
> > > We sincerely thank you for acknowledging that all concerns have been fully resolved, and for the positive and constructive engagement throughout the review process.
> > >
> > > We would like to respectfully highlight two additional points for consideration:
> > >
> > > 1. **Community enablement.** We have open-sourced the key innovation implementation and training pipeline and released the largest real e-commerce search dataset currently available (10× existing scale), which we hope lowers the barrier for future research in this direction. Due to the double-blind policy, we are unable to provide direct links at this stage.
> > >
> > > 2. **External adoption.** The COINS paper (WWW 2026) has independently adopted our RQ-OPQ encoding and reported significant improvements, suggesting our technical designs have broader applicability beyond our specific deployment.
> > >
> > > We are grateful for your positive assessment. If these points provide additional perspective on the contribution scope, we would appreciate any reconsideration of the score. Thank you again for the valuable review.

---

### Decision · Program_Chairs · 2026-04-30

**Decision:**

Accept (regular)

**Comment:**

- This paper addresses an important and practically challenging problem in industrial ML by proposing a unified end‑to‑end generative framework to replace traditional multi‑stage cascading architectures in large‑scale e‑commerce search. The technical soundness of the system design and the practical significance of unifying retrieval and ranking under strict latency and scalability constraints are generally agreed on.

- Several initial concerns—particularly regarding reproducibility, deployment feasibility, and the role of the reward model during inference—were clarified during the rebuttal. The authors’ responses, including clarification of deployment details and the release of core implementation and an anonymized large‑scale dataset, substantially improved reviewer confidence and led to a score increase from weak accept to accept in one case.

- Overall, despite some remaining limitations in evaluation clarity and comparison to academic generative retrieval baselines, the work is technically sound and offers meaningful practical contributions by bridging generative retrieval approaches with real‑world production‑scale search deployment.

- For these reasons, I recommend acceptance.